# Root exudate metabolites drive plant-soil feedbacks on growth and defense by shaping the rhizosphere microbiota

Lingfei Hu [1], Christelle A.M. Robert[1], Selma Cadot[2], Xi Zhang[1], Meng Ye [1], Beibei Li[1], Daniele Manzo[1], Noemie Chervet[3], Thomas Steinger[3], Marcel G.A. van der Heijden[2,4,5], Klaus Schlaeppi [1,2] & Matthias Erb [1]

By changing soil properties, plants can modify their growth environment. Although the soil microbiota is known to play a key role in the resulting plant-soil feedbacks, the proximal mechanisms underlying this phenomenon remain unknown. We found that benzoxazinoids, a class of defensive secondary metabolites that are released by roots of cereals such as wheat and maize, alter root-associated fungal and bacterial communities, decrease plant growth, increase jasmonate signaling and plant defenses, and suppress herbivore performance in the next plant generation. Complementation experiments demonstrate that the benzoxazinoid breakdown product 6-methoxy-benzoxazolin-2-one (MBOA), which accumulates in the soil during the conditioning phase, is both sufficient and necessary to trigger the observed phenotypic changes. Sterilization, fungal and bacterial profiling and complementation experiments reveal that MBOA acts indirectly by altering root-associated microbiota. Our results reveal a mechanism by which plants determine the composition of rhizosphere microbiota, plant performance and plant-herbivore interactions of the next generation.

[1] Institute of Plant Sciences, University of Bern, 3013 Bern, Switzerland. [2] Division of Agroecology and Environment, Agroscope, 8046 Zürich, Switzerland. [3] Division of Plant Protection, Agroscope, 1260 Nyon, Switzerland. [4] Department of Evolutionary Biology and Environmental Studies, University of Zürich, 8057 Zürich, Switzerland. [5] Institute of Environmental Biology, Utrecht University, 3508 TC Utrecht, The Netherlands. These authors contributed equally: Klaus Schlaeppi, Matthias Erb. Correspondence and requests for materials should be addressed to K.S. (email: klaus.schlaeppi@ips.unibe.ch) or to M.E. (email: matthias.erb@ips.unibe.ch)

Many organisms have the capacity to influence the local environment and thereby modify individual performance. Plants for instance change the soil environment and thereby modulate local growth conditions for themselves, as well as for other plants, including their own offspring. So-called plant-soil feedbacks are important determinants of plant succession[1], plant population, community structure[2], and plant diversity[3]. In agriculture, plant-soil feedbacks are exploited in the form of crop rotation, where the sequence of crops is adjusted to provide optimal soil conditions for crop yield and environmental sustainability[4].

Plant-soil feedbacks are often connected to changes in soil biota. The composition of soil microbial communities in particular has been recognized as an important player in this context[5]. The accumulation of pathogens for instance can suppress plant growth, while the accumulation of beneficials such as nitrogen-fixing bacteria or mycorrhizal fungi can improve plant performance. While the effects of a few individual microbes on plant performance are well studied, the broader contribution of entire soil microbiomes to plant-soil feedbacks remains to be uncovered.

Plants can change the soil microbiota by secreting bioactive molecules into the rhizosphere. Root exudates typically comprise primary metabolites such as sugars, amino acids, and carboxylic acids, as well as a diverse set of secondary metabolites[6–9]. Besides representing carbon and nitrogen substrates for microbial growth, root exudate compounds have a multitude of effects on rhizosphere microbes by acting as signaling molecules, attractants, stimulants, but also as inhibitors or repellents[10]. Thereby, the composition of root exudates, which is under host-genetic control, likely defines the assembly of plant-specific root and rhizosphere microbial communities[11]. The exudation of bioactive metabolites varies substantially between different plant species, as do their microbial communities and plant-soil feedback effects[12]. Thus, root exudate metabolites may drive plant-soil feedbacks by modifying microbial communities. However, clear evidence for this hypothesis is currently lacking, and the proximal mechanisms underlying plant-soil feedbacks remain unknown.

Maize root exudates contain a variety of metabolites, including substantial amounts of benzoxazinoids (BXs) that are secreted to the rhizosphere[13,14]. BXs present bioactive indole-3-glycerol-phosphate derived secondary metabolites that are found across the *Poaceae*. BXs have been studied extensively as important herbivore and pathogen resistance factors. Although, BXs have not yet been investigated in the context of plant root and rhizosphere microbiomes, they are known to trigger rhizosphere colonization by the plant-growth promoting bacterium *Pseudomonas putida*[15] or inhibit host recognition and virulence of the pathogenic *Agrobacterium tumefaciens*[16].

Here, we investigated the impact of BX exudation on microbial community composition and the resulting effects on plant growth and defense as a potential mechanism underlying plant-soil feedbacks. Using a BX-deficient maize mutant, we first tested whether BXs structure root-associated microbial communities, and whether compositional microbiota changes are associated with changes in plant growth, defense, and herbivore resistance of the next plant generation. To evaluate resistance, we used *Spodoptera frugiperda*, an economically important and invasive leaf pest of maize[17]. We then pharmacologically complemented soils with BXs to establish a causal link between BX accumulation in soil and soil-feedbacks. We also sterilized conditioned soils and complemented them with microbial extracts to causally link the BX-dependent changes in soil microbiota to the observed BX-dependent soil-feedback effects of the next plant generation. In summary, we present a mechanism explaining how plants condition the rhizosphere microbiota by exuding bioactive molecules and thereby affect growth and defense of the next plant generation.

## Results

**Benzoxazinoids determine root-associated microbiota.** To test if BXs alter the root-associated microbiota, we first grew wild type (WT) B73 plants and a near-isogenic line of a *bx1* maize mutant[18] in the field. *Bx1* encodes a tryptophan synthase alpha homolog that produces indole as a BX precursor. By consequence, the *bx1* mutant is BX deficient[18]. Root surface washes revealed that *bx1* plants exude 90% less BXs than WT plants (Fig. 1a, Supplementary Fig. 1a), and extracts of soil cores contained significantly less BXs after 3 months of plant cultivation (Fig. 1b, Supplementary Fig. 1b, c). Three months after planting at the end of the vegetative growth stage, we performed bacterial[19] and fungal[20] ribosomal marker gene profiling of soil, root, and rhizosphere samples of the field-grown WT and *bx1* plants (Supplementary Movie 1, Supplementary Data 1–4). We did not observe any effects on α-diversity (refer to the Supplementary Data 3 for in depth analysis of BX exudation effects on microbial diversity). Both unconstrained and constrained ordinations revealed that community composition of bacteria and fungi differed markedly between WT and *bx1* plants (Fig. 1c, d, Supplementary Fig. 1d, e). Multivariate statistics confirmed the significance of these effects (Supplementary Tables 1 and 2). Closer inspection of individual operational taxonomic units (OTUs) revealed substantial and compartment-specific microbiota variation between B73 and *bx1* plants (Supplementary Fig. 1f–k). *Bx1*-dependent effects accounted for ~5% of total microbiota variation (Supplementary Table 1), a level of variation that is typically observed in multi-loci comparisons between different varieties or accessions[21]. Thus, the differential exudation of BXs from maize roots is associated with marked changes in the microbiota on the roots and in the rhizosphere.

**Soil conditioning by benzoxazinoids increases plant defense.** To test whether the BX-dependent variation in rhizosphere microbiota composition is associated with changes in plant performance, we measured growth and herbivore defense of maize plants growing in soil cores previously conditioned by WT or *bx1* mutant plants, referred to as 'BX+' and 'BX−' soils, respectively. Plants growing in BX+ soil were smaller, contained less chlorophyll, and accumulated less shoot biomass than plants growing in BX− soil (Fig. 2a, Supplementary Fig. 2a, b). Furthermore, *Spodoptera frugiperda* caterpillar growth was reduced on leaves of plants grown in BX+ soil (Fig. 2b, Supplementary Fig. 2c). The amount of leaf damage did not differ between soil types (Fig. 2c), indicating that the leaves were less palatable rather than less attractive for the caterpillars. Metabolite profiling revealed that leaves of plants growing in BX+ soil contained reduced levels of sugars, hydrolysable amino acids, and total soluble protein, but increased levels of the defense signals salicylic acid (SA), jasmonic acid (JA), and 12-oxo-phytodienoic acid (OPDA; Fig. 2d, Supplementary Fig. 2d, e). Defensive marker genes such as a JA-responsive serine proteinase inhibitor (*ZmSerPIN*) and the toxic ribosome-inactivating protein 2 (*ZmRIP2*)[22,23] were also upregulated (Fig. 2d, Supplementary Fig. 2f). Concentrations of defensive metabolites, as well as defensive marker genes such as the SA-marker *ZmPR1* and the proteinase inhibitor *ZmMPI*[22] on the other hand did not differ between soil treatments (Fig. 2d, Supplementary Fig. 2f, g). Together, this reveals that soil conditioning by BX exudation is associated with increased expression of JA-responsive defenses and herbivore growth suppression, as well as decreased primary metabolite accumulation and plant growth in the next plant generation.

**Effects of benzoxazinoids on growth are genotype specific.** To corroborate the involvement of the BX pathway in the observed soil-conditioning effects on plant performance, we conditioned

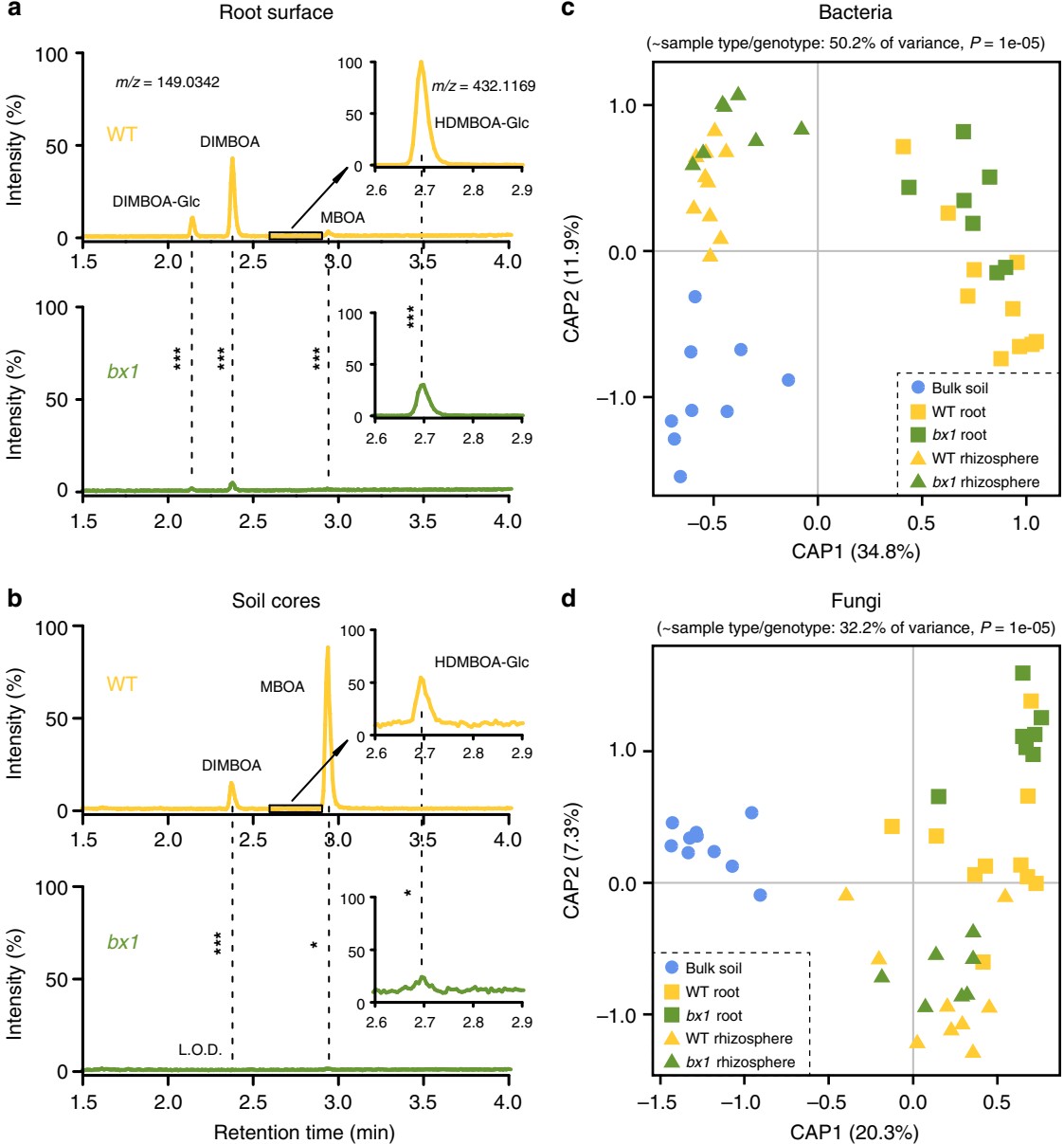

**Fig. 1** Benzoxazinoid release by maize roots is associated with changes in root-associated microbiota. **a**, **b** Benzoxazinoid (BX) profiles at the root surface (**a**, $n = 14$–15) and the soil cores (**b**, $n = 5$) of wild type (WT) B73 and *bx1* mutant plants. For absolute quantities, refer to Supplementary Fig. 1. Stars indicate significant differences (two-sided Student's *t* tests, *$P < 0.05$; ***$P < 0.001$). **c**, **d** Partial Canonical analysis of Principal Coordinates (CAP) of rhizosphere and root bacterial (**c**) and fungal (**d**) communities ($n = 7$–10). The CAP ordinations using Bray–Curtis distance were constrained for nesting the factor plant genotype within the factor sample type. Model, explained fraction of total variance and model significance are indicated above the plots. Axes report the proportions of total variation explained by the constrained axes. Non-conditioned bulk soil samples were included as negative controls. For unconstrained ordination and individual operational taxonomic units (OTUs), refer to Supplementary Fig. 1

soils with three additional *bx* mutants and their respective wild type line W22[24,25] in the greenhouse and determined plant and herbivore performance in the next plant generation. Consistent with the findings in the B73 background, herbivore growth decreased on W22 plants growing in W22-conditioned soil compared to *bx1*, *bx2* or *bx6* mutant-conditioned soils (Fig. 3a). However, no biomass effect was observed in the W22 background (Fig. 3b). Physiological measures corroborated these differential phenotypes. Coherent with the enhanced herbivore resistance (Fig. 3a) and similar to B73 plants, the response plants in the W22 background exhibited increased levels of the defense hormones and defense marker genes (Fig. 3c, Supplementary Fig. 3). In contrast to B73 and consistent with the lack of a growth effect

in W22 plants (Fig. 3b), the leaves of W22 response plants did not differ in soluble protein levels irrespective of previous conditioning (Fig. 3c, Supplementary Fig. 3). Thus, BX-dependent increases in leaf defenses appear to be genetically uncoupled from growth suppression, and the growth effect of BX soil conditioning varies with the plant genetic background.

**Benzoxazinoid effects outlast a winter fallow period.** In an agricultural context, soil conditioning and growth of the next plant generation are typically separated by a fallow period. To account for this effect, we conditioned soils in the field, left them to overwinter in the field and then assessed these BX+ and BX− soils for feedbacks on plant growth and defense during the next

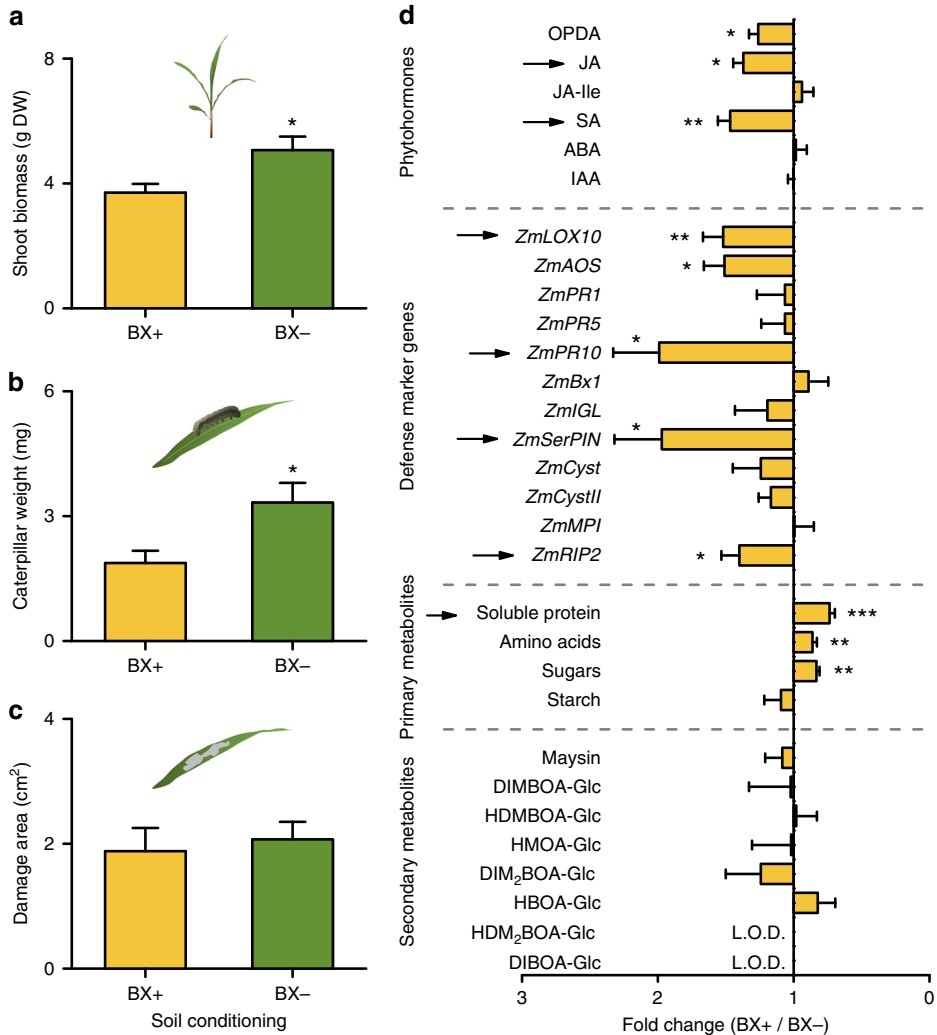

**Fig. 2** Benzoxazinoid soil conditioning increases plant defense and decreases plant growth. **a** Shoot biomass of 10-week old wild type (WT) B73 plants growing in soils previously conditioned by WT (BX+) or benzoxazinoid (BX)-deficient *bx1* mutant plants (BX−) (*n* = 10). **b**, **c** Weight gain (**b**) and leaf damage (**c**) of *Spodoptera frugiperda* caterpillars (*n* = 10). **d** Changes in leaf phytohormones, defense marker genes, primary and secondary metabolites (*n* = 9-10). Stars indicate significant differences between conditioning treatments (two-sided Student's *t* tests, *$P < 0.05$; **$P < 0.01$; ***$P < 0.001$). Arrows indicate metabolic markers which were used for phenotyping in subsequent experiments. DW, dry weight. L.O.D., below limit of detection

vegetation period. The effects of BX soil conditioning on plant and herbivore performance occurred independently of the fallow period (Supplementary Fig. 4).

**The breakdown product MBOA is sufficient for soil-feedbacks.** Six-methoxy-benzoxazolin-2-one (MBOA) was identified as the dominant *Bx1*-dependent metabolite in BX+ soils (Fig. 1b, Supplementary Fig. 1b). We therefore hypothesized that this metabolite may be sufficient to trigger the observed changes in plant growth and defense. To test this hypothesis, we measured total MBOA levels in soils of individual pot-grown WT plants over 3 months (Supplementary Fig. 5a) and supplemented the soils of *bx1* plants with the matching amounts of MBOA over the same period of time. MBOA complementation restored the growth and herbivore phenotypes of the subsequent plant generation (Fig. 4a, b), and reverted BX-responsive leaf metabolic markers back to WT levels (Fig. 4c, Supplementary Fig. 5b–h). Together, these results demonstrate that soil-derived MBOA is both sufficient and necessary to increase defense and decrease growth of the next plant generation.

**Changes in soil microbiota trigger feedback effects.** Root exudates may influence the performance of the next plant generation through a series of factors, including changes in pH, nutrients, soil structure, and soil biota[26]. Elemental analysis and pH measurements did not reveal any significant differences between BX+ and BX− soils at the end of the conditioning phase (Supplementary Table 3) and after the response phase (Supplementary Table 4). Also, mixing conditioned field soils with 50% unconditioned potting soil with high organic matter and nutrient content did not affect the feedback effects in the next plant generation (Supplementary Fig. 6). We therefore evaluated if BX-dependent changes in soil microbiota may be responsible for the observed feedback effects. To test this hypothesis, we sterilized BX+ and BX− soils through X-radiation and complemented the sterilized soils with microbial extracts of BX+ and BX− soils. Microbial extracts were obtained through 25 μm filtration[27], resulting in solutions which contained soil microbes, but no members of the larger soil macrofauna. Soil sterilization eliminated the differences in plant and herbivore performance and metabolic leaf markers between BX+ and BX− soils (Fig. 5, Supplementary Fig. 7, Supplementary Tables 5, 6).

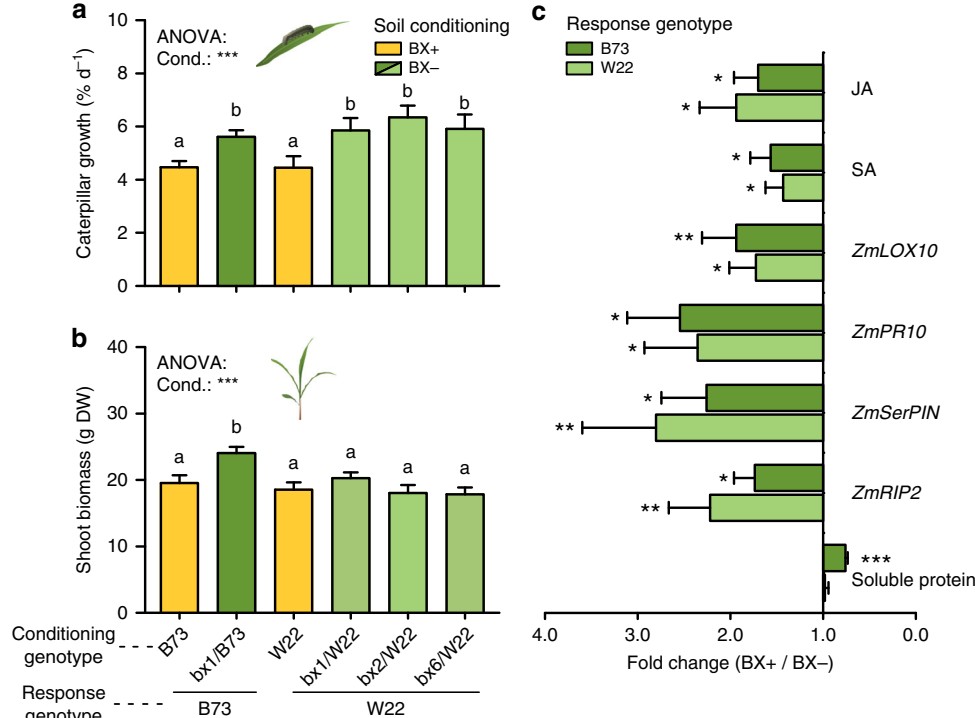

**Fig. 3** Influence of soil conditioning of different benzoxazinoid mutants on plant and herbivore performance. **a**, **b** Average caterpillar weight gain (**a**) and shoot biomass (**b**) of the next generation of wild type (WT) B73 or W22 maize plants ("Response genotype") growing in soils previously conditioned by corresponding WT (BX+) or benzoxazinoid (BX)-deficient *bx*-mutant plants (BX−) ("Conditioning genotype", +SE, $n = 10$–15). Stars indicate significant differences between soil types (***$P < 0.001$, one-way ANOVA). Different letters indicate significant differences between individual soil types ($P < 0.05$, one-way ANOVA followed by multiple comparisons through FDR-corrected LSMeans). **c** Fold changes of leaf markers in the next generation of B73 or W22 plants growing in BX+ soil which were conditioned by corresponding WT (B73 or W22) plants relative to those growing in BX− soil which were conditioned by corresponding *bx*-mutant plants ($n = 10$). Stars indicate significant differences between soil types (ANOVA, FDR-corrected LSMeans, *$P < 0.05$; **$P < 0.01$; *** $P < 0.001$). For full datasets, refer to Supplementary Fig. 3. DW, dry weight. Cond., conditioning

Complementation with microbial extracts of BX+ and BX− soils fully restored the BX-dependent effects on plant growth and defense (Fig. 5, Supplementary Fig. 7, Supplementary Tables 5, 6).

**Relationship between MBOA, microbiota, and feedback effects**. To investigate the connection between the soil microbiota and MBOA, we complemented *bx1* plants with physiological levels of MBOA (Supplementary Fig. 5a) and then sterilized a subset of control and MBOA-treated soil cores with X-radiation. MBOA complementation without subsequent sterilization reduced plant and caterpillar growth and induced leaf defenses similar to the previous experiment (Fig. 6, Supplementary Fig. 8a–g). These effects disappeared when MBOA-complemented soils were sterilized (Fig. 6, Supplementary Fig. 8a–g). X-radiation did not influence residual MBOA concentrations in the soil (Supplementary Fig. 8h), which allows us to exclude direct effects of soil-derived MBOA on plant growth and defense. Together, these results show that MBOA-induced changes in soil microbiota are sufficient to explain BX-dependent effects on plant growth and defense.

**Benzoxazinoids shape microbiota of the next plant generation**. To understand whether BX soil conditioning changes the root-associated microbiota of the responding plants, we profiled fungal and bacterial communities in root and rhizosphere samples of WT plants grown in BX+ and BX− soils. Both unconstrained and constrained ordinations revealed a clear separation of bacteria and fungi between rhizosphere and roots and between BX+ and BX− soils (Fig. 7a, b, Supplementary Fig. 9a, b). Multivariate

statistics confirmed the significance of these effects (Supplementary Tables 7 and 8; we refer to the Supplementary Data 3 for in depth analysis of BX soil conditioning effects on microbial diversity). Actinobacteria OTUs and a subset of OTUs belonging to Ascomycota and Glomeromycota contributed most strongly to the separation of root and rhizosphere samples (Fig. 7c, d). A group of Proteobacteria was more abundant in BX+ soils while some Chloroflexi OTUs together with OTUs of other phyla were characteristic for BX− soils. Fungal communities were more variable and exhibited a less clear taxonomic pattern with subsets of Ascomycota OTUs found for either type of soil conditioning. Additionally, we noted that Glomeromycota OTUs tended to be less abundant in BX+ soils, suggesting they are negatively affected by BX conditioning. While OTU-level analysis confirmed variation for specific bacterial OTUs between root and rhizosphere samples of BX+ and BX− soils, this was not the case for fungal OTUs due to heterogeneous community composition (Supplementary Fig. 9c–h, see Supplementary Data 4). Thus, microbiota-mediated BX-dependent effects on plant growth and defense are more strongly associated with changes in bacteria than fungi in the rhizosphere of the responding plants.

## Discussion

Plants are well known to modify soil microbiota and thereby determine the performance of their offspring[5], but the proximal mechanisms underlying this phenomenon are not well understood. Here, we describe a mechanism through which maize plants determine growth and defense of the next generation by changing the composition of bacterial and fungal communities in

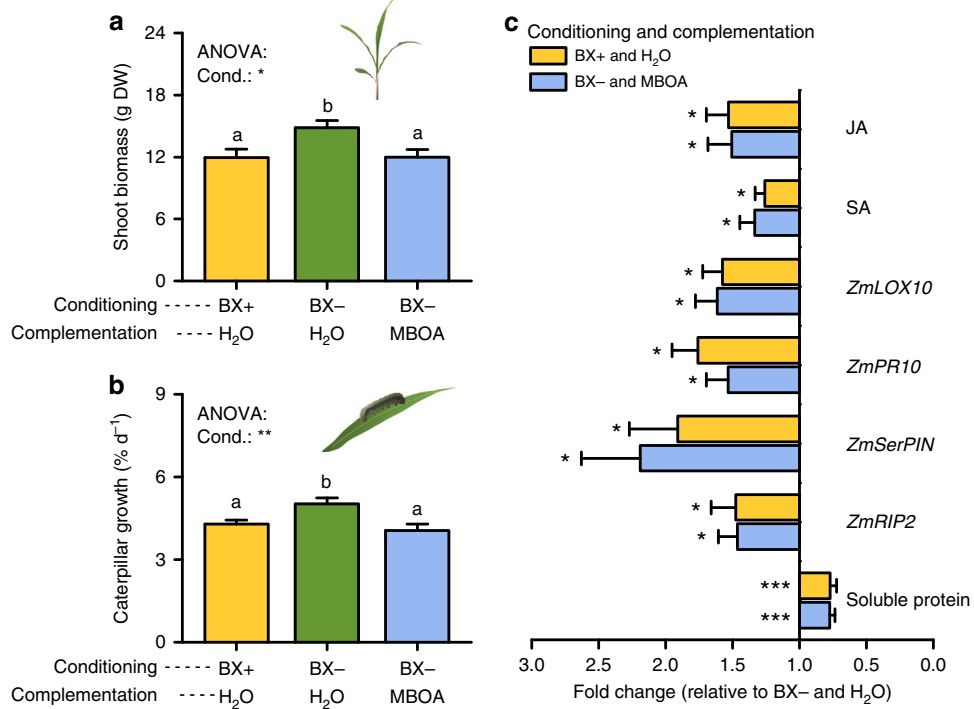

**Fig. 4** Application of MBOA restores benzoxazinoid-dependent plant-soil feedback effects. **a**, **b** Shoot biomass (**a**) and caterpillar growth rate (**b**) of wild type (WT) B73 plants growing in soils previously conditioned by WT (BX+) or benzoxazinoid (BX)-deficient *bx1* mutant plants (BX−), as well as MBOA-complemented BX− soils ($n = 15$–19). Letters indicate significant differences between conditioning treatments (ANOVA, FDR-corrected LSMeans, $P < 0.05$). **c** Fold changes of leaf markers in WT B73 plants growing in BX+ soil or MBOA-complemented BX− soil relative to water-treated BX− soils ($n = 15$–19). For full datasets, refer to Supplementary Fig. 5. No significant differences were found between BX+ soil and BX− soil supplemented with MBOA (Supplementary Fig. 5). DW, dry weight. Cond., conditioning. Stars indicate significant differences between treatments and BX− soil (ANOVA, FDR-corrected LSMeans, *$P < 0.05$; **$P < 0.01$; ***$P < 0.001$)

the soil. Our findings support the following mechanistic model (Fig. 8): Maize plants release a blend of metabolites, including BXs such as DIMBOA, from the roots and thereby influence the composition of the root-associated microbiota (Fig. 1). DIMBOA is relatively short-lived and is rapidly and spontaneously converted into the more stable MBOA[28], which accumulates in the soil (Fig. 1). Soil conditioning by MBOA, and possibly other BXs, triggers changes in the structure of the root-associated microbiota in the next plant generation (Fig. 7). These changes increase leaf defenses, suppress herbivore growth and decrease plant growth, the latter depending on the plant genetic background (Fig. 2). Future experiments are needed to clarify the exact nature of the BX-dependent factors that are transmitted from the conditioning to the response phase. Complementation experiments suggests that larger members of the soil fauna are not required for the feedback effects[27]. Instead, microbes and/or their metabolites are likely to be transmitted and may determine the assembly of the microbiota in the next generation[29]. Detailed microbiome analyses, including approaches that reduce the noise generated by relic DNA in the soil[30], as well as high-resolution metabolite fingerprinting combined with activity screens of the microbial extracts from BX+ and BX− conditioned soils will help to test these hypotheses.

The suppression of herbivore growth in B73 plants growing in BX+ soils is associated with an increase in leaf concentrations of the defensive phytohormones SA and JA and stronger expression of JA, but not SA-responsive defense marker genes (Fig. 2, Supplementary Fig. 2). Note that *ZmPR10* in maize is JA rather than SA responsive[22]. JA-dependent defenses are well established to increase maize resistance against herbivores, including *Spodoptera frugiperda*[31]. Although SA can suppress JA signaling in many

plants[32], it has been shown to prime JA-dependent defenses in maize[33]. Therefore, it is likely that the BX-dependent changes in soil microbiota increase SA and JA signaling in the leaves and thereby trigger JA-dependent defenses that subsequently induce enhanced resistance to herbivores. The fact that SA marker genes were not induced despite the enhanced SA levels points to a negative regulation of SA signaling by JA[34]. To a certain degree, the observed phenotypes are reminiscent of induced systemic resistance (ISR), a phenomenon where individual root-colonizing rhizobacteria increase pathogen and herbivore resistance in plant leaves[35]. ISR acts by promoting JA signaling in various plant species including Arabidopsis, rice, and maize[35–37]. The rhizobacterium *Pseudomonas putida* strain KT2440, which is attracted to maize BXs[15] primes systemic, JA-dependent defense responses[38] and triggers ISR against the maize anthracnose fungus *Colletotrichum graminicola*[37]. Although ISR is traditionally associated with single rhizobacteria interacting with a plant, we expect complex microbial communities to accommodate the same traits and to be capable to elicit ISR. Our study represents an example where a specific microbiota composition is associated with improved plant defenses. Future work is required to clarify the role and capacity of microbiomes to elicit ISR. Such work will contribute to a more general understanding of how the plant microbiota supports the host immune system[39].

Apart from enhanced defenses, B73 plants growing in BX+ soils displayed reduced shoot growth, leaf amino acids, total soluble protein, and sugars. The negative growth effect and total protein depletion were absent in the W22 background. This result shows the growth and defense effects of BX+ conditioning can be uncoupled in W22, which has potential implications for agriculture. Maize, which is often grown in the same field for several

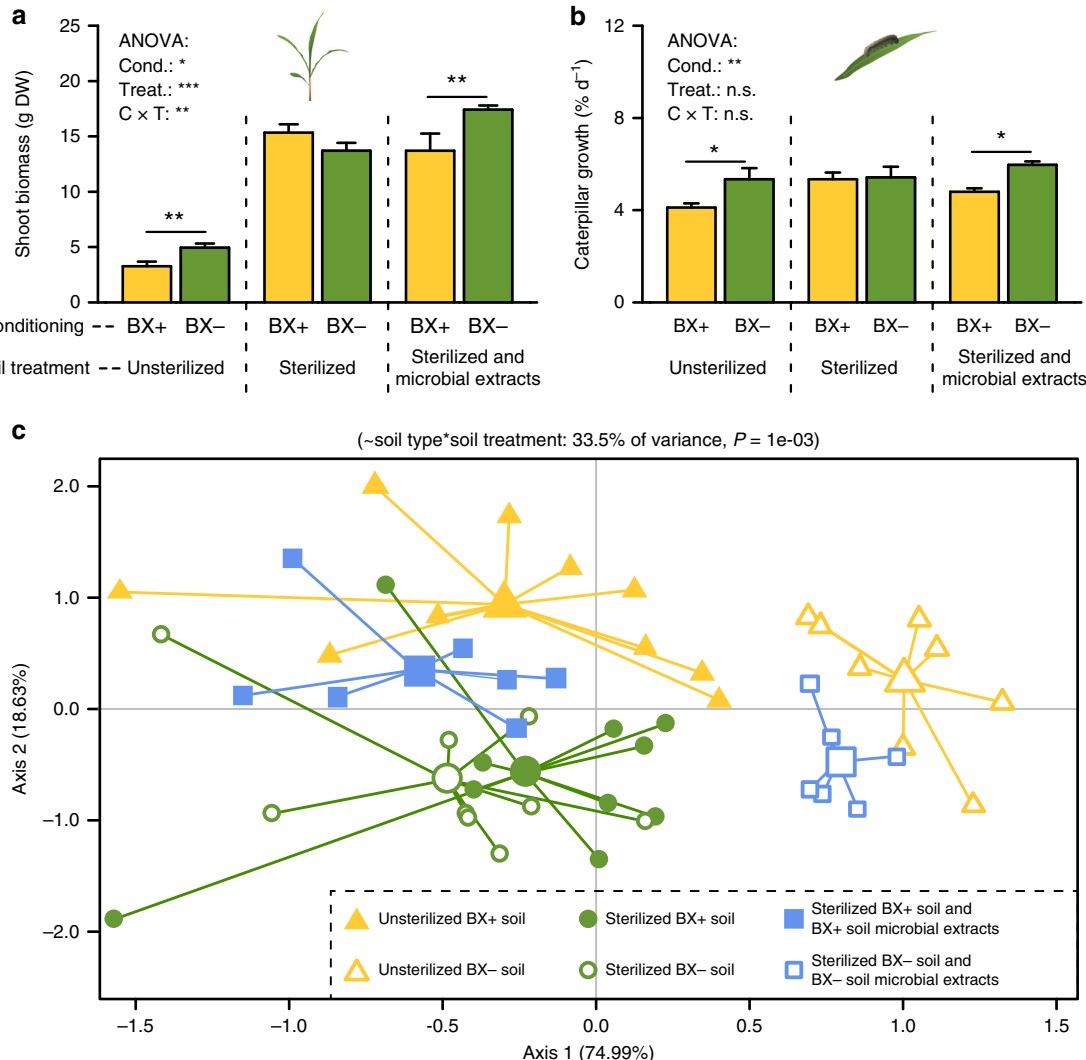

**Fig. 5** Benzoxazinoid-dependent changes in soil microbiota are necessary and sufficient to trigger changes in plant and herbivore performance. **a**, **b** Plant (**a**) and herbivore performance (**b**) of wild type (WT) B73 plants growing in soils previously conditioned by WT (BX+) or benzoxazinoid (BX)-deficient *bx1* mutant plants (BX−). Soils were either left untreated, sterilized by X-ray or sterilized and complemented with microbial extracts from the respective non-sterilized soils. Stars indicate significant differences within soil treatments (ANOVA, FDR-corrected LSMeans, *$P < 0.05$; **$P < 0.01$; $n = 6$–10). **c** Redundancy analysis (RDA) of leaf markers ($n = 6$–10). Model, explained fraction of total variance and model significance are indicated above the plots. Axes report the proportions of constrained variation explained by the constrained axes. For corresponding unconstrained ordination and individual metabolite measures, refer to Supplementary Fig. 7 and Supplementary Tables 5 and 6. DW, dry weight. Cond., conditioning. n.s., no significant. Treat., treatment

consecutive years[40], may benefit from BX soil conditioning and microbiome modulation through enhanced defense and herbivore resistance[41]. Depending on the maize genotype however, growth and yield may be reduced by BX soil conditioning. Although often considered as relatively self-tolerant, maize is well known to benefit from crop rotation[40]. Our work suggests that farmers may be able to further enhance self-tolerance in maize by planting genotypes that do not display negative plant-soil feedbacks. Understanding the genetic architecture of BX-dependent growth suppression, both in terms of potential direct allelopathic effects[42] and indirect microbiome-mediated soil feedback effects, is an exciting prospect in this context.

Genomes of plant varieties or accessions vary by different alleles at multiple loci, and this high level of host genetic diversity typically explains ~5–6% of compositional variation in root and rhizosphere microbiota[21]. A similar level of heritable variation had also been identified among of 27 maize inbred lines[43] (genomes of inbred lines differ at multiple loci). Strikingly, we find the

same level of microbiota variation (Supplementary Table 1) by comparing plants that are genetically nearly identical but differ in BX production. This observation suggests that the BX pathway plays a major role in structuring the maize root and rhizosphere microbiota. Future experiments could expand these findings to include different soil types and rotation scenarios with different response crops, to evaluate whether the observed microbiome responses and effects are widespread and of importance for sustainable cropping systems. It will also be interesting to investigate if and how BXs interact with other bioactive exudate metabolites from maize such as flavonoids[44], which may also shape the maize rhizosphere microbiota.

Root exudation fuels the substrate-driven assembly process of the plant-specific root and rhizosphere microbiota from the surrounding soil biome[11]. While this overall function of root exudates is undisputed, it is not well understood how specific compounds in complex root exudates influence microbial community structure. Microbial communities are known to respond

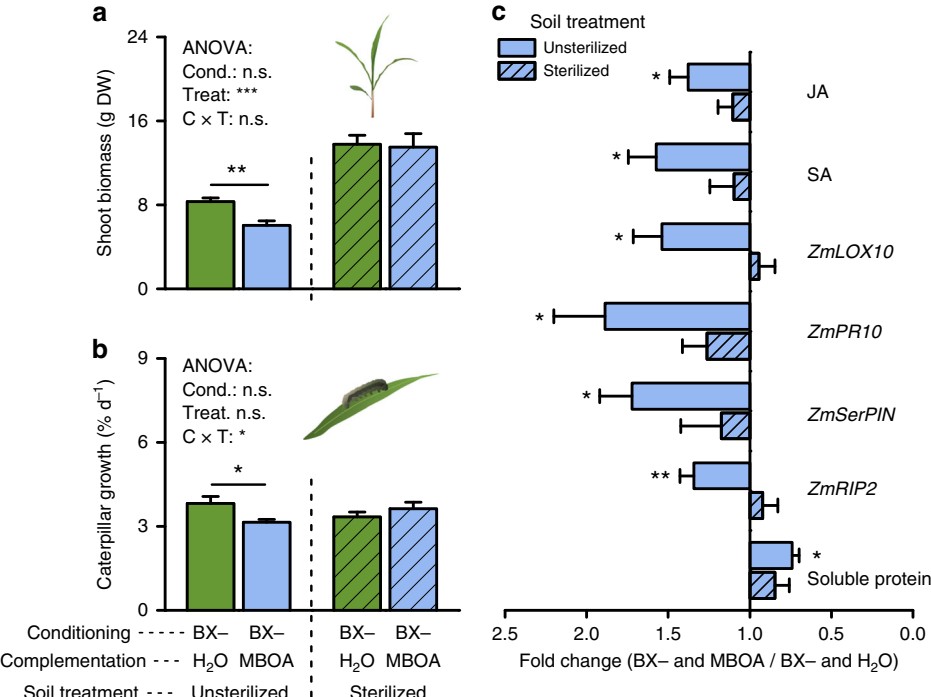

**Fig. 6** MBOA-induced effects on plant and herbivore performance depend on the soil microbiota. **a**, **b** Shoot biomass (**a**) and caterpillar growth rate (**b**) of wild type (WT) B73 plants growing in soils previously conditioned by benzoxazinoid (BX)-deficient *bx1* mutant plants (BX−), which were complemented with water or MBOA with and without subsequent sterilization (+SE, *n* = 11). **c** Fold changes of leaf markers in WT B73 plants growing in BX− soils which were complemented with MBOA relative to water-treated BX− soils with and without subsequent sterilization (+SE, *n* = 11). For full datasets, refer to Supplementary Fig. 8. Cond. or C, conditioning. Treat or T, treatment; n.s., no significant; DW, dry weight. Stars indicate significant differences between complementation treatments within soil treatments (ANOVA, FDR-corrected LSMeans, *\*P* < 0.05; *\*\*P* < 0.01; *\*\*\*P* < 0.001)

dynamically to available nutrients and chemical environments[45]. For instance, specific taxa of the soil bacteria community increased in abundance in a carbon substrate-dependent manner to individual additions of common root exudate compounds such as glucose, glycine, and citric acid[46]. Here, we expand this view using a genetic approach by showing that in addition to primary metabolites, secondary metabolites such BXs have a major effect on microbial community structure and microbiome traits. The presented study thus represents a step towards understanding plant chemistry driven plant-microbiome interactions.

Interactions between plants and soils determine the coexistence, succession, and productivity of plants in natural and agricultural systems[3,26,47,48]. This study establishes that plants influence growth and herbivore defense of the next generation of plants by altering the soil microbiota through the secretion of bioactive molecules from the roots. This finding extends the current view on the importance of heritable plant traits in modulating plant-associated microbiomes[43,49–53] by establishing a pivotal role of exuded plant secondary metabolites. Furthermore, the experiments provide a functional link between exudate-dependent changes in soil microbiota and plant performance. Given that the release of bioactive metabolites into the rhizosphere is a common feature in plants[10], microbiome-mediated effects of root exudates on plant performance are likely to be widespread in natural and agricultural ecosystems.

## Methods

**Plant and insect resources**. The maize (*Zea mays* L.) genotypes B73, *bx1*/B73 (referred to as *bx1*), W22, *bx1*/W22, *bx2*/W22, and *bx6*/W22 were used in this study. For a detailed description of these genotypes, see[18,24,25]. *Spodoptera frugiperda* (J.E. Smith, 1791) larvae were reared on artificial diet as described by Maag et al.[54].

**Field experiment**. We planted B73 and *bx1* maize plants in an arable field at Agroscope in Changins, Switzerland (Parcel 30, 2014; 46°23′56.7′′ N, 6°13′58.9′′ E). The field was managed according to conventional Swiss farming practices, including the use of agrochemicals, with a 6-year rotation and a cropping history of alfalfa (2013), rapeseed and alfalfa (2012), winter wheat (2011), soybean (2010), winter wheat (2009), and maize in 2008. Individual plants were randomly distributed and separated by at least 3 buffer plants within rows, and rows with test plants were mixed with rows of buffer plants (hybrid variety Delprim). After 3 months, we excavated soil cores of approximately 20 × 20 × 20 cm containing the root system from individual B73 or *bx1* maize plants and separated the root system from the soil cores for microbiota determination. The bulk soil from each plant was collected for feedback experiments and stored at 4 °C until use.

**Collection of samples for microbiota profiling**. From 10 randomly chosen root systems, we sampled a 10 cm fragment corresponding to a soil depth between 5 and 15 cm (Supplementary Movie 1) into a fresh Petri dish using ethanol-sterilized scissors. We cut the fragment into 3 cm pieces and transferred them into 50 ml Falcon tubes. The roots were washed 4 times with 25 ml of sterile ddH$_2$O, shaking the tubes 10 times vigorously at each step. The wash fractions were combined with centrifugation steps (5 min at 3220×*g*, discarding the supernatant) and the resulting pellets were defining the rhizosphere samples and stored at −20 °C until further use. The washed roots were transferred to 15 ml Falcon tubes, lyophilized for 72 h and subsequently milled to a fine powder using a Retsch Ball Mill (Retsch GmBH, model MM301; settings 30 s at 30 Hz using one 1-cm steel ball). These samples were used for microbiota analysis as described below. As the root sampling method does not discriminate between the inner root tissue and the root surface, we refer to the sampling unit as "root microbiota".

**Feedback experiment I**. The soil samples from the soil cores of the different plants from the field experiment were individually passed through a 1 cm sieve, homogenized, mixed with 20% autoclaved sand, and used to fill 3 L pots (13.1 cm depth and 20.2 cm diameter). New B73 plants were individually grown in pots containing B73-conditioned or *bx1*-conditioned soil (*n* = 9–10). Pots were randomly placed on a greenhouse table (26 °C ± 2 °C, 55% relative humidity, 14:10 h light/dark, 50,000 lm m$^{-2}$) and re-arranged weekly. Plants were watered three times per week.

Seventy days after planting, plant height and chlorophyll content of the youngest fully opened leaf of each plant were recorded. Chlorophyll contents were determined using a SPAD-502 meter (Minolta Camera Co., Japan). The youngest fully developed leaf of each plant was harvested and divided into two parts. Half of

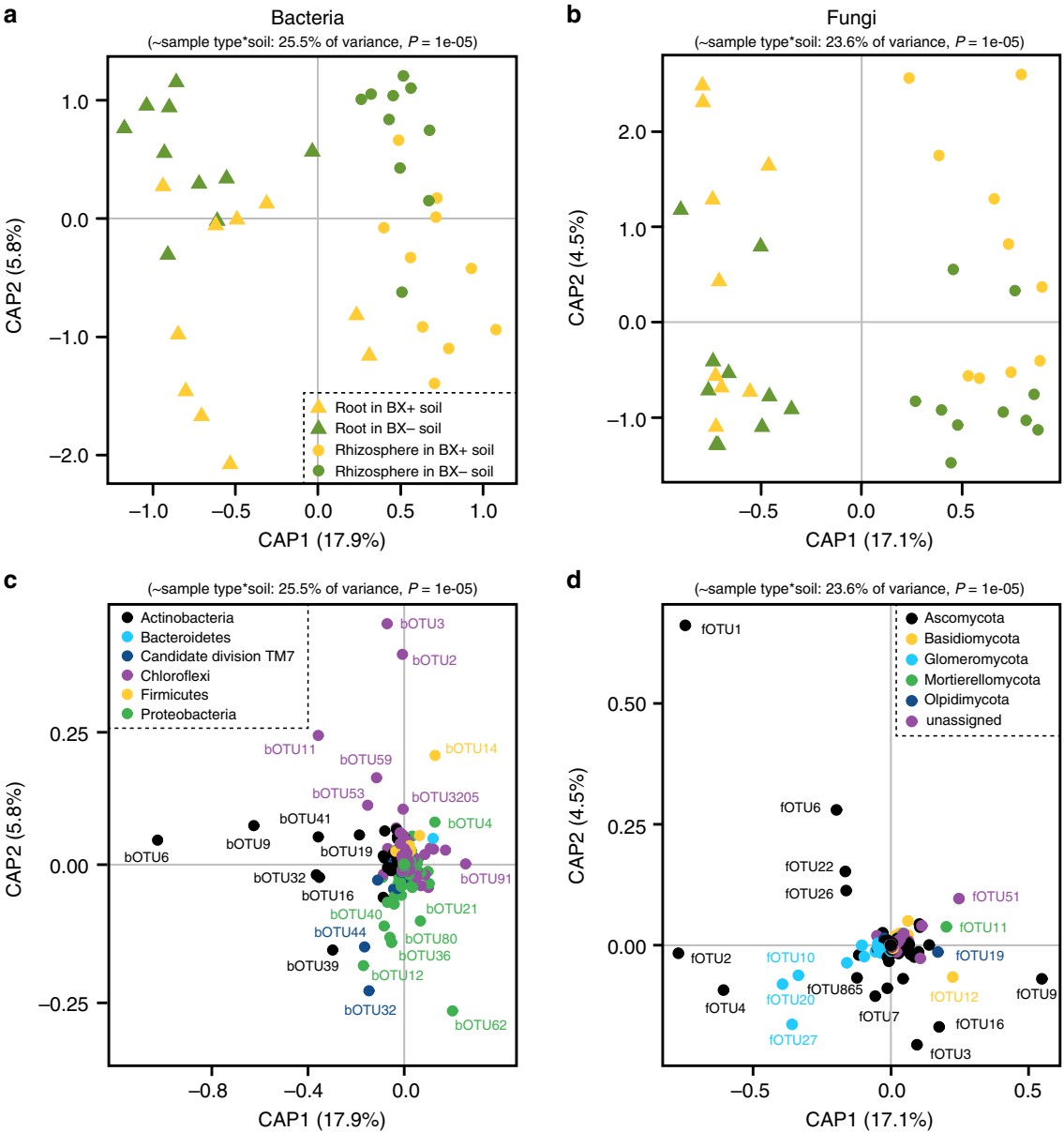

**Fig. 7** Benzoxazinoid-conditioning shapes soil microbiota in the roots and rhizosphere of the second plant generation. **a**, **b** Partial Canonical analysis of Principal Coordinates (CAP) of rhizosphere and root bacterial (**a**) and fungal (**b**) communities of wild type (WT) plants grown in soils previously conditioned by WT (BX+) or benzoxazinoid (BX)-deficient *bx1* mutant plants (BX−) (*n* = 10). CAP ordinations using Bray–Curtis distance were constrained for the factors sample type and soil conditioning. Model, explained fraction of totasl variance and model significance are indicated above the plots. Axes report the proportions of total variation explained by the constrained axes. **c**, **d** Biplot with the bacterial Operational Taxonomic Unit (bOTU, **c**) and fungal OTU (fOTU, **d**) scores of the CAP showing the contribution of individual OTUs to the separation of the different treatments. For unconstrained ordination and individual OTUs, refer to Supplementary Fig. 9

it was used to measure *S. frugiperda* performance (see below). The other half of the youngest fully developed leaf was immediately frozen in liquid nitrogen and stored at −80 °C for analyses of primary and secondary metabolites, as well as defensive markers (see below). After 91 days of growth, the experiment was stopped, the remaining above-ground biomass was harvested, oven dried at 60 °C for 2 days and weighed. Root systems were removed from the pots, root and rhizosphere samples were collected as described above for microbiota analysis.

**Herbivore growth assays.** Five neonates were starved for 2 h and then placed into transparent solo cups (4 cm height and 3.5 cm diameter). Halves of the youngest fully developed leaves of individual maize plants were placed into the cups. Larval mass of *S. frugiperda* was determined 5–7 days after the start of the experiment. This method was used for different experiments as described above (*n* > 8). To quantify damage, the remaining leaf pieces were scanned, and the removed leaf area was quantified using Digimizer 4.6.1 (Digimizer). We also determined larval performance on intact plants. One starved and pre-weighted second instar lara was

introduced into a cylindrical mesh cage (1 cm height and 2.5 cm diameter), which was clipped onto the youngest fully developed leaf of individual maize plants. The position of the cages was moved to provide sufficient food supply for the larvae. Larval mass was recorded 5 days after the start of the experiment (*n* = 10).

**Microbial complementation, soil structure, and benzoxazinoids.** To explore the role of soil microbiota and soil structure in the feedback effects, B73 and *bx1* maize plants were grown again in the field (2015; neighboring field plot (46°23′47.2″N, 6° 14′28.9″E), same agricultural management). The soil cores from individual plants were collected after 3 months, sieved and homogenized as described above, divided into three parts, and used for three different experiments.

For the first experiment (microbial complementation), the soil cores were further divided into five sets of aliquots. The first and second sets of aliquots were sterilized by X-ray (25 kGy minimum to 60 kGy maximum) at Synergy Health Däniken AG, Switzerland. The third set of aliquots was used to determine MBOA concentrations after X-ray sterilization as described below. The fourth set of

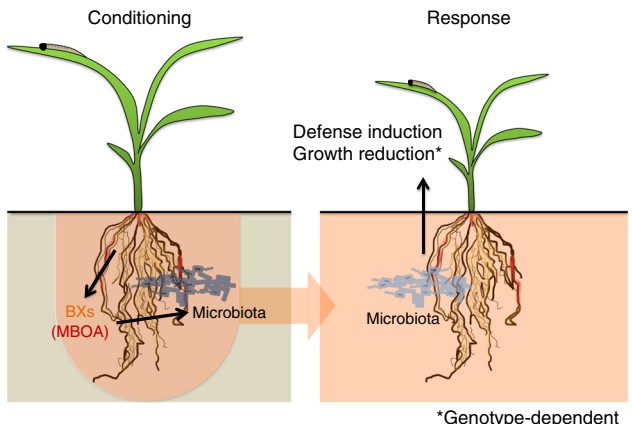

**Fig. 8** Proposed model for benzoxazinoid-dependent, microbiota-mediated plant performance

aliquots was used to obtain microbial extracts and complement one of the sterilized aliquots. Briefly, 500 mL portions of soil were mixed thoroughly with 500 mL autoclaved Milli-Q water. The mixtures were left to stand for 2 h to let large soil particles settle. Then the supernatants were sieved through a 250 μm sieve followed by two sieves of 25 μm, which can retain nematodes, spores of most species of arbuscular mycorrhiza while letting the suspended microorganisms pass through. Two hundred milliliter of microbial extracts were used for complementation. A fifth set of soil aliquots was left untreated and used as a positive control. The different sets of sterilized and unsterilized soil aliquots were individually mixed with 20% autoclaved sand, and filled into 1 L pots (13.5 cm depth and 11.7 cm diameter). Half of the pots containing sterilized soil were then complemented with the microbial extracts, resulting in a total of 6 soil types. Wild type B73 plants were then planted into the different pots and phenotyped as described ($n = 6$–10). Analyses included plant shoot biomass, *S. frugiperda* performance and the expression of a subset of seven conditioning-responsive metabolites and defense markers which were identified in the first feedback experiment (see Fig. 2). The same phenotypes and markers were analyzed in all subsequent experiments unless described otherwise (see below).

For the second experiment (soil structure), the field soils were divided into two sets of aliquots. Both aliquots were mixed with 20% autoclaved sand. In addition, one set of aliquots was mixed with 50% commercial greenhouse potting soil (Klasmann–Deilmann GmbH, Germany). After thorough mixing, these soils were filled into 1 L pots (13.5 cm depth and 11.7 cm diameter). B73 plants were individually planted in the pots, and placed in the greenhouse and phenotyped ($n = 9$–10).

For the third experiment (quantification of benzoxazinoids in soil), five soil cores of previously grown B73 and *bx1* maize plants were extracted and analyzed for benzoxazinoids (see below).

**MBOA complementation experiment.** Fresh bulk soil from the first field site was collected in 2015 and filled into 3 L pots (13.1 cm depth and 20.2 cm diameter). To condition the bulk field soil, B73 and *bx1* plants were planted and grown in a common garden in Bern (46°57′10.6′′ N, 7°26′41.0′′ E). Plants were watered with different solutions once a week: B73 plants were watered with water only, while half the *bx1* plants were watered with water only and the other half with MBOA (dissolved in water). In order to determine the MBOA concentration to be used for application to the soil of *bx1* plants, an additional set of B73 plants was grown in 200 mL pots using the same field soil one week ahead of the experiment. MBOA was extracted from an individual subset of the pots every week and quantified by UHPLC-UV as described below ($n = 4$). The determined amounts of MBOA per plant (Supplementary Fig. 5a) were then used for the weekly complementation of the *bx1* plants (refer to values in Supplementary Fig. 5a). After 3 months, the conditioned soil of each plant was harvested separately, sieved, homogenized, and divided into three sets of aliquots. The first set of aliquots was mixed with 20% autoclaved sand and filled into 1 L pots (13.5 cm depth and 11.7 cm diameter). New B73 plants were individually grown in each pot in the greenhouse and phenotyped after 70 days as described above ($n = 15$–19). The other two sets of aliquots were used for a second experiment. One set of aliquots was sterilized as described above, and the other set was left untreated. Both sterilized and unsterilized soils were mixed with 20% autoclaved sand and filled into 1 L pots (13.5 cm depth and 11.7 cm diameter). B73 plants were planted into the pots in the greenhouse and phenotyped after 70 days ($n = 11$).

**Benzoxazinoid pathway experiments.** Fresh bulk field soil from the field in Changins was sieved, homogenized and filled into 3 L pots (13.1 cm depth and 20.2 cm diameter). Then, B73, *bx1*/B73, W22, *bx1*/W22, *bx2*/W22, and *bx6*/W22 seeds were planted in these pots in the common garden in Bern. After 3 months, the

conditioned bulk soil of each plant was harvested, sieved, homogenized, mixed with 20% autoclaved sand and filled into a 1 L pot (13.5 cm depth and 11.7 cm diameter). New wild type plants were then grown in the different pots in the greenhouse and phenotyped after 70 days ($n = 10$–15).

**Feedback persistence experiment.** To explore the persistence of BX-dependent soil feedback effects, B73 and *bx1* maize plants were grown in a field experiment in Changins (2015; neighboring field plot (46°23′47.2″N, 6°14′28.9″E), same agricultural management). After 3 months of conditioning, the shoots of the plants were cut, and the fallow field was left to overwinter. During the next growing season, the soil cores from individual plants were collected, sieved, homogenized, mixed with 20% autoclaved sand, and used to fill 3 L pots (13.1 cm depth and 20.2 cm diameter). New B73 plants were then grown in the different pots in the greenhouse and phenotyped after 70 days ($n = 11$–13).

**Analysis of benzoxazinoids, AMPO, AAMPO, and maysin.** To quantify the defensive metabolites in the plants from the soil feedback experiment, 60 mg samples of flash frozen and ground maize leaves were extracted in 600 μL of acidified $H_2O$/MeOH (50:50 v/v; 0.1% formic acid), and analyzed with an Acquity UHPLC-MS system equipped with an electrospray source (Waters i-Class UHPLC-QDA, USA; $n = 9$–10). Compounds were separated on an Acquity BEH C18 column (2.1 × 100 mm i.d., 1.7 μm particle size). Water (0.1% formic acid) and acetonitrile (0.1% formic acid) were employed as mobile phases A and B. The elution profile was: 0–9.65 min, 97-83.6% A in B; 9.65-13 min, 100% B; 13.1–15 min 97% A in B. The mobile phase flow rate was 0.4 mL/min. The column temperature was maintained at 40 °C, and the injection volume was 5 μL. The MS was operated in negative mode, and data were acquired in scan range ($m/z$ 150–650) using a cone voltage of 10 V. HDMBOA-Glc and DIMBOA-Glc were quantified in positive mode using single ion monitoring (SIM) at $m/z$ 194 with cone voltage of 20 V. All other MS parameters were left at their default values as suggested by the manufacturer.

To determine the levels of benzoxazinoids on the root surface, B73 and *bx1* seeds were pinned on needles (5 cm length), and germinated in aluminum containers (21.5 cm length, 14.5 width, and 6.5 depth) at 100% humidity. This allowed the seeds to germinate without direct contact to substrate and their exudates to be sampled in vivo without damaging the root system. One week after germination, 2 cm root sections of individual intact seedlings were rinsed by 50 μL Milli-Q water ($n = 14$–15). Then, 50 μL MeOH (0.2% formic acid) were immediately added to obtain a final solution of $H_2O$/MeOH (50:50 v/v; 0.1% formic acid).

To determine the concentration of BXs and BX breakdown products in field soil, 300 mL B73-conditioned or *bx1*-conditioned bulk soils (see microbial complementation and soil structure experiments) were completely homogenized in 200 mL acidified $H_2O$/MeOH (50:50 v/v; 0.1% formic acid). The resulting mixture was filtered and the remaining soil was discarded. The liquid phase was centrifuged two times for 10 min at 16,200×g, and the supernatant was recovered ($n = 5$).

To explore whether soil sterilization can cause MBOA degradation, MBOA ($n = 6$) was extracted from 300 mL unsterilized and sterilized B73-conditioned field soil (see microbial complementation and soil structure experiments) using the same procedure.

Benzoxazinoids on the root surface and in the field soil were quantified using an Acquity UHPLC system coupled to a G2-XS QTOF mass spectrometer (MS) equipped with an electrospray source (Waters, USA) as described[55]. AMPO and AAMPO were quantified by using the following modifications. The elution profile was: 0–3.5 min, 99–72.5% A in B; 3.50–5.50 min, 72.5–50% A in B; 5.51–6.5 min 100% B, 6.51–7.51 and 99% A in B. The QTOF-MS was operated in positive mode. Capillary and cone voltages were set to 3.07 kV and 6 V, respectively.

To quantify the amount of MBOA over time (see MBOA complementation experiment) soils were collected, homogenized with 200 mL $H_2O$/MeOH (50:50 v/v; 0.1% formic acid), filtered, and centrifuged as described above. The recovered supernatants were concentrated to 2 mL by freeze drying (Swiss Vacuum Technology SA, Switzerland), and then injected into a UHPLC-UV system (Waters, USA). MBOA was quantified by measuring absorbance at 285 nm.

In all experiments, absolute quantities of secondary metabolites were determined using standard curves obtained from purified DIMBOA, MBOA, DIMBOA-Glc, HDMBOA-Glc, AMPO, AAMPO, and maysin as described[56,57].

**Extraction and analysis of primary metabolites.** To evaluate the feedback effects on primary metabolites of the succeeding maize plants, we measured the contents of hydrolyzable amino acids, soluble sugars, soluble protein and starch in the youngest fully opened leaf of each plant from the soil feedback experiment. Amino acids were hydrolyzed and quantified by UHPLC-MS (Waters, USA) according to a previously published protocol[58]. Soluble sugars (glucose, sucrose, and fructose) ($n = 7$–10) were extracted and quantified as described[59]. Starch concentrations ($n = 7$–9) were determined using a Starch (GO/P) Assay Kit (Amylase/Amyloglucosidase method, Sigma, USA) following the manufacturer's instructions. Soluble protein was extracted and quantified for plants from different experiments using a Bradford assay[60], this method was used for different experiments as described above.

**Quantification of phytohormones**. Maize leaves were harvested and ground in liquid nitrogen as described as above. The phytohormones OPDA, JA, JA-Ile, SA, IAA, and ABA were extracted with ethyl acetate spiked with isotopically labeled standards (1 ng for $d_5$-JA, $d_6$-ABA, $d_6$-SA, $^{13}C_6$ -JA-Ile, and 100 ng for $d_5$-IAA) and quantified by UHPLC-MS-MS as described in a previous study[61].

**Expression of defense marker genes**. Quantitative real time PCR (QRT-PCR) was used to measure the expression of defense marker genes. Total RNA was isolated from ground leaves using the GeneJET Plant RNA Purification Kit (Thermo Scientific, USA) following the manufacturer's instructions. One microgram of each total RNA sample was reverse transcribed with SuperScript® II Reverse Transcriptase (Invitrogen, USA). The QRT-PCR assay was performed on the LightCycler® 96 Instrument (Roche, Switzerland) using the KAPA SYBR FAST qPCR Master Mix (Kapa Biosystems, USA). A linear standard curve, threshold cycle number versus log (designated transcript level), was constructed using a serial dilution of cDNA, and the relative transcript levels of the target genes in the samples were determined according to the standard curve. The maize actin gene *ZmActin1*[22] was used as an internal standard to normalize cDNA concentrations.

**Soil analyses**. Chemical and physical parameters were measured from 1 kg sub-samples of the soils, which were collected after the growth of B73 or *bx1* from the field experiment. Soil texture and pH, total C and N, macronutrients and micro-nutrients were extracted with 1:10 ammonia-acetate-EDTA and determined according to the reference methods of the Swiss Federal Research Stations (Eid-genoessische Forschungsanstalten FAL, RAC, FAW, 1996)[62]. In addition, we measured pH, soil texture and macronutrients in the soil at the end of the response phase. Soil parameters are listed in Supplementary Tables 3 and 4.

**Microbiota profiling**. Approximately 200 mg of fine root powder, rhizosphere or soil sample were employed as input for DNA extraction with the FastDNA® SPIN Kit for Soil (MP Biomedicals, USA) following the manufacturer's instructions. DNA concentrations of all samples were determined on a Varian Eclipse Fluor-escence plate reader (Agilent, USA) using Quant-iT$^{TM}$ PicoGreen® dsDNA Assay Kit (Invitrogen, USA) and Herring Sperm DNA (Invitrogen, USA) as standard solution.

We largely followed Hartman et al. (2017) for bacteria profiling based on 16 S rRNA gene sequencing[63]. Briefly, the V5-V7 regions of the 16 S rRNA gene were amplified with barcoded PCR primers 799 F and 1193 R. The Supplementary Data 1 contains the mapping of samples and the utilized barcodes. Triplicate PCR reactions consisted of 1 × 5Prime Hot Mastermix (5Prime, USA), 0.3% Bovine Serum Albumin (New England Biolabs, UK), 400 nM of barcoded primers (Sigma Aldrich, Switzerland), and 10 ng template DNA in a total reaction volume of 20 μL. Cycling conditions consisted of 2 min at 94 °C, 30 cycles of 30 s at 94 °C, 30 s at 55 °C, and 30 s at 65 °C, and 10 min at 65 °C. PCR products were validated for correct size and absence of contamination by gel electrophoresis, followed by gel purification with the NucleoSpin Gel and PCR cleanup kit (Macherey–Nagel, Germany) and DNA quantification. Gel purification is required to remove the plastid derived PCR product (light gel band at approx. 800 bp) that is co-amplified by the PCR primers 799 F and 1193 R on root samples.

We profiled the fungi similar to McGuire et al. (2013)[20] based on the first internal transcribed spacer (ITS) region using the PCR primers ITS1F and ITS2. Consistent with the bacteria profiling, both the forward and reverse primers a padding sequence and an 8-bp error-correcting unique barcode (5'-pad-barcode-primer-3', see Supplementary Data 1). Triplicate PCR reactions consisted of 1 × 5Prime Hot Mastermix (5Prime, USA), 0.3% Bovine Serum Albumin (New England Biolabs, UK), 200 nM of barcoded primers (Sigma Aldrich, Switzerland), and 5 ng template DNA in a total reaction volume of 20 μL. Cycling conditions consisted of 3 min at 94 °C, 30 cycles of 45 s at 94 °C, 60 s at 50 °C, and 90 s at 72 °C, and 10 min at 72 °C. PCR products were pooled and validated for correct size and absence of contamination by gel electrophoresis and followed by DNA quantification.

Sequencing libraries were assembled by pooling 50 ng of each sample followed by a purification step with AMPure XP beads (Beckman Coulter, USA) and DNA quantification. Sequencing adapters were ligated to the library by the Functional Genomics Center Zurich (http://www.fgcz.ch/) followed by sequencing on a MiSeq instrument in paired-end 2 × 300 bp mode (Illumina). Separate bacteria libraries had been prepared and sequenced separately for the field and the soil feedback experiments. The ITS library contained both experiments and was sequenced in a third run.

The raw sequencing data is available from the European Nucleotide Archive (http://www.ebi.ac.uk/ena). The bacteria of the field experiment samples were sequenced in a MiSeq run together with clover experiments (presented in Hartman et al. (2017)[63] and are available under the study accession PRJEB15152 (Sample: SAMEA54297418)). The MiSeq runs containing the bacteria profiles of the feedback experiment, as well as the fungi profiles of both experiments are available under the study accession PRJEB20127. The bioinformatic processing included a quality filtering of the raw sequencing read data using PRINSEQ[64], merging with FLASH v.1.2.9[65] and de-multiplexing with Cutadapt[66]. Quality

sequences were trimmed to a fixed length of 360 bp, sorted by abundance, de-replicated, and clustered to operational taxonomic units (OTUs, ≥ 97% sequence similarity, minimal coverage of 5 sequences) with UPARSE v8.1.1812[67]. Chimeric OTU sequences were removed using the built-in chimera detection tool of UPARSE[68,69]. Taxonomy assignments were performed using the SILVA 16 s v119 database[70] and the UNITE database[71] (dynamic release 28.06.2017) with the RDP classifier implemented in QIIME v1.8[72]. The bioinformatics script including all individual parameters and support files used are provided as Supplementary Data 2.

**Statistical analysis**. Data was analyzed by analysis of variance (ANOVA) after confirming that it met the assumptions of normality and equal variance, and then followed by pairwise or multiple comparisons of Least Squares Means (LSMeans), which were corrected using the False Discovery Rate (FDR) method[73]. To compare the metabolic profiles of different treatments, we used redundancy analysis (RDA) as described previously[74]. Significant differences between treatments were determined by Monte Carlo tests with 999 permutations. The above analyses were conducted using R 3.2.2[75] using the packages "car", "lsmeans", "vegan" and "RVAideMemoire"[76-79]. Microbiota profiles were analyzed employing the packages 'vegan', 'sciplot', 'coin', 'phyloseq' and 'edgeR'. Microbiota profiles were filtered to exclude OTUs classified as eukaryotes, Cyanobacteria or assigned to mito-chondria. The entire microbiota and statistic analysis including all parameters and settings is encoded in an R-markdown file and is available together with all input files required for replication of the analysis as Supplementary Data 3. The Supplementary Data 3 also comprises PDF reports (the R-markdown output) that summarizes the bacteria and fungi analyses in R and provides additional background information, descriptions of the sequencing effort, as well as explanations and justifications for the analysis logic. Briefly, because we found for our sampling groups significant differences between the mean sequencing depths (Kruskal-Wallis test, $P < 0.05$), we rarefied the OTU table as recommended by Weiss et al. (2015)[80]. They demonstrated that rarefying removes artifacts due to group-wise differences in sequencing depths better than other normalization techniques. The rarefied OTU table was utilized for diversity analyses and identification of differentially abundant OTUs ($P$-values were adjusted for false discovery rate (FDR) according to Benjamini and Hochberg[73]. For visualization, we expressed the OTU abundances as percentages of the total number of counts in a sample. Further statistical details can be found in Supplementary Data 4.

**Experimental considerations and sample sizes**. Sample sizes were chosen based on previous experience within this system, and experiments were fully randomized. Analyses were blinded by assigning numbers instead of treatment labels to individual samples and tracing back treatment assignments after data collection. No samples were excluded during data analysis. Most sample sizes can be found in the figure legends or are directly visible from PCA and RDA plots. Additional individual sample sizes are as follows: Fig. 1a, WT, $n = 14$, $bx1$, $n = 15$; Fig. 2d, BX+, $n = 9$, BX−, $n = 10$; Fig. 3a, b, B73, $n = 15$, $bx1$/B73, $n = 12$, W22, $n = 14$, $bx1$/W22, $n = 10$, $bx2$/W22, $n = 14$, $bx6$/W22, $n = 14$; Fig. 4a–c, BX+ and $H_2O$, $n = 14$, BX− and $H_2O$, $n = 19$, BX− and $H_2O$, $n = 17$.

**Data availability**. The raw sequencing data of microbiota is available through the European Nucleotide Archive (PRJEB15152 [Sample: SAMEA54297418, https://www.ebi.ac.uk/ena/data/view/SAMEA54297418] and PRJEB20127, [https://www.ebi.ac.uk/ena/data/view/PRJEB20127]). The sequence data of maize genes can be found in the GenBank/EMBL database under the following accession numbers *ZmActin1* (MZEACT1G, [https://www.ncbi.nlm.nih.gov/nuccore/168403]), *ZmBx1* (AY254103, [https://www.ncbi.nlm.nih.gov/nuccore/AY254103]), *ZmIGL* (AF271383, [https://www.ncbi.nlm.nih.gov/nuccore/AF271383]), *ZmLOX10* (DQ335768, [https://www.ncbi.nlm.nih.gov/nuccore/DQ335768]), *ZmAOS* (AY488135, [https://www.ncbi.nlm.nih.gov/nuccore/AY488135]), *ZmRIP2* (L26305, [https://www.ncbi.nlm.nih.gov/nuccore/L26305]), *ZmMPI* (X78988, [https://www.ncbi.nlm.nih.gov/nuccore/X78988]), *ZmCyst* (CK371502, [https://www.ncbi.nlm.nih.gov/nucest/CK371502]), *ZmCystII* (D38130, [https://www.ncbi.nlm.nih.gov/nuccore/D38130]), *ZmSerPIN* (BM382058, [https://www.ncbi.nlm.nih.gov/nucest/BM382058]), *ZmPR1* (U82200, [https://www.ncbi.nlm.nih.gov/nuccore/U82200]), *ZmPR5* (U82201, [https://www.ncbi.nlm.nih.gov/nuccore/U82201]), *ZmPR10* (AY953127, [https://www.ncbi.nlm.nih.gov/nuccore/AY953127]). All relevant data supporting the findings of this study are available from the corresponding authors [M.E. and K.S.] upon request.

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

## Acknowledgements

We thank Martijn Bezemer, Davide Bulgarelli, and Georg Jander for insightful discussions and input on an earlier version of the manuscript. The Agroscope soil laboratory (Switzerland) performed soil nutrition and structure analysis. Inge S. Fomsgaard (Aarhus University, Denmark) and Gaétan Glauser (University of Neuchâtel, Switzerland) provided analytical standards. This project was supported by ERA-CAPS (BENZEX) the Swiss State Secretariat for Education, Research and Innovation (Project C15.0111), the Swiss Sino Science and Technology Exchange Programme (EG 03-032016), the Swiss National Science Foundation (Nr. 136184, 160786, 164480, and 165891) and the Interfaculty Research Cooperation "One Health" of the University of Bern.

## Author contributions

L.H. conceived, designed, performed, and analyzed experiments and wrote the first draft of the manuscript. C.A.M.R. conceived, designed, performed, and supervised experiments. S.C., X.Z., M.Y., B.L., and D.M. performed experiments. N.C. designed and performed experiments. T.S. designed and supervised experiments. M.G.A.v.d.H. supervised experiments. K.S. conceived the study, designed, performed, supervised, and analyzed experiments. M.E. conceived the study, designed, performed, supervised, and analyzed experiments and wrote the first draft of the manuscript. All authors contributed to the final version of the paper.
