## [Peer Review File · Nature Communications]

Reviewers' comments:

Reviewer #1 (Remarks to the Author):

Hu et al describe a series of experiments that show quite convincingly that the root exudate MBOA, which accumulates in the soil after maize is grown there, causes a shift in the soil microbiota that in turn impacts the growth and level of defense of the next generation of maize. This understanding is achieved with a combination of plant mutants that do not produce benzoxazinoids (which breaks down to produce MBOA), soil sterilization experiments to isolate the critical role of the microbiota, MBOA addition to isolate the importance of that particular exudate, following experiments to assess agricultural importance, microbiota characterization to show plausibility and experiments using a second plant background. I believe that they have made a solid case for a novel finding. The paper is well written, concise and clear, and a pleasure to read.

There were a few things that I would have liked to see considered, although none of them rise to the importance of impacting my overall positive impression of the work.

1. There is an argument about extended phenotypes in the summary. We do know, however, that there can be substantial variation in the microbiota associated with particular plant genotypes. The most comprehensive examination was in Nature Communications by Horton et al. (2014: 5:5320 doi: 10.1038/ncomms6320) and is curiously not cited in this paper. It seems that the impact on the microbiota achieved by the W22 genotype impacts the B73 genotype (although not its own biomass). This doesn't seem consistent with the idea of an extended phenotype. How does heritable variation in the microbiome impact this interpretation of your results? What might be the importance of individual differences in the microbiome?
2. It is observed that W22 does not share a benefit of modifying its microbiome as was shown for B73. What might drive this difference? It would be interesting to map host variation in this ability!
3. The importance of the increases in SA and JA seem to warrant some discussion. First, since there is well known crosstalk then why are both increasing? Second, do I get it right that these hormones increase in concentration with no impact on downstream triggering of the pathway? If so, why does resistance to herbivory increase?
4. Are the compositional changes in OTU relative abundances consistent across the two plant host genotypes?

Reviewer #2 (Remarks to the Author):

Hu et al demonstrate 1 week old bx1 seedlings roots produce less BXs compared to wt plants. They also demonstrate that field soil contains detectable levels of BX1s by 3 month old wt plants, while soil from 3 month old bx1 plant BXs are not detectable. They hypothesize the BXs in the soil may influence the microbiome, which is consistent with previous studies demonstrating BX can have negative and positive impacts on microbes. Using the same site they collect root, rhizosphere and bulk soil (3-month-old wt and bx1 plants) and examine bacterial and fungal communities using 16s and ITS sequencing. They demonstrate the soil, rhizosphere, and root microbiota are clearly differed from each other, the differences between wt and bx1 samples were not as clear.

The authors hypothesized that BXs induced changes in soil may influence the next crop planted in the soil. To test this soil from the same site was taken to the lab/greenhouse and wt plants were grown in both types (bx1 conditioned soil and wt conditioned soil). They observed plants were bigger (Dry Weight- Was water content different), had more nutrients, and fewer defenses in leaf tissue when wt plants were grown in soil from bx1 plots compared wt plots. This translated to more palatable plants

for herbivores as evidenced in bigger caterpillars on plants grown in bx1 soil compared to wt soil. Taken together these results suggest bx1 plants influenced component of the soil at this site that are important for plant health and defense.

The authors next try to demonstrate which BXs is driving changes in soil by performing a complementation experiment, (although they only examine the impact of 1 BX that was changing in the soil). They take soil from the same field the next year and condition it in the lab with wt and bx1 plants and then grow wt plants in the soil from both treatments. In half the plants growing in bx1 soil they add the BX MBOA. This 'rescues' the wild type soil phenotype (reduced insect growth, reduced plant growth, increased defenses, decreased nutrients) suggesting MBOA is the BX driving changes among treatments. Finally, they try to pin down the component of soil health the BXs are influencing at this site. Using chemical analysis and a series of subtractive experiments (homogenizing soil and sterilizing soil) they demonstrate microbes are important for changes in plant in bx1 soils.

Nice paper and a hard work. There are many studies demonstrating BXs can have negative and positive impacts on microbes and on the relationship of BXs in induction of plant defense. This study does a nice job taking it one step further demonstrating at this site BX1 induced changes in microbes-plant interactions and this influenced plant defense potential to above ground herbivores.

Questions and concerns:

I wonder how the results would differ if soil was taken from another site with a different microbiota? Or if potting soil (without field soil) was used in the lab and conditioned with wt and bx1 plants before being used in the experiments? I expect the impact on plant defense in the next generation largely depends on what microbes are present.

Maize is typically rotated as was done at the authors field site (6-year rotation stated by authors). It is not clear if BXs would still be in the soil after crop rotation or if changes in the soil microbiome would still be evident after rotation and able to influence the next generation of maize? It seems the next cover crop or the cover crop immediately prior to planting maize again will likely have a larger influence on the microbiome, compared to BX mediated changes in microbes several years before.

Could using another plant species or genotype that also alters the soil microbiome result in decreased plant defense in the next generation? Maybe it is not specific to the BXs, just changes in microbiome? The complementation experiment suggests specificity but discussing the previous studies would clarify how this study is different. Previous studies have shown crop rotation can alter nutrients and increase disease incidence and plant susceptibility in the following generation. Changes in/influence of microbiome have also been measured in some of these systems.

Why are BX concentration presented so different from figure to figure? (Expressed relative to cm² of root, per L, per pot). It is hard to compare across experiments the relative concentrations observed and used. Can you standardize the presentation so it is easier to compare across experiments? Maybe this is not possible.

The bx1 conditioned field soil data suggests no BXs were present at the site from previous maize propagation – so MBOA is not stable for that long? What about the presence of BXs in the conditioned soil before and after each experiment? This would strengthen the connection and potential for impact on the next generation. Is always MBOA the highest BX in the soil? Other BXs were changing significantly in the conditioned field soil (for example - DIMBOA, HDMBOA-glc, HBOA-glc). Could these also be driving changes?

The W22 genotype experiment was nice – but plant defenses and nutrients were not measured in the plants growing in conditioned soils. Also there was no impact on plant growth for w22 experiment, although insect growth was still increased in the bx1/w22. Is this also due to a lack of defense or could it be due to only increased nutrient availability? This experiment should be done if it was not already.

It would have been nice if the authors also conditioned potting soil or some other soil with bx1 and wt and demonstrated the same findings. It may be very specific finding to this one site and set of microbes and the authors should address this with experiments or in the discussion.

Before the lab experiments the field 'conditioned' soil was analyzed for changes in chemistry/health. No changes in soil chemistry/health were observed in the field collected samples. Could soil chemistry/health be changing over the 3 months of the greenhouse/lab experiments? The microbe populations could still be driving the changes. This would tell you if changes in soil chemistry from the microbes or direct plant-microbe interactions are important for the phenotype in the next generation.

Legend missing on extended figure 1 16s and ITS data.

Reviewer #3 (Remarks to the Author):

Nature Communication manuscript NCOMMS-17-25861-T

Comments to the Authors

As the major claim, the manuscript deals with the question whether benzoxazinoids can function as modifiers of plant microbiome composition resulting in impacts on the next plant generation.

Regarding benzoxazinoids comparable studies have not been done, but effects of other secondary metabolites on subsequent cultures are known. The idea that plant secondary metabolite triggered microbiomes induce systemic resistance against pathogens and herbivores is not new. Such influences are important for agriculture and they are therefore of general interest.

In detail, the authors investigated possible influences of MBOA on the soil and root microbiota, subsequent impacts on maize and growth of *Spodoptera frugiperda* caterpillars feeding on leaves of these maize plants. They used BX 73 or mutants as soil conditioner in comparison to non-conditioned field soils and performed complementation experiments with MBOA. They found a decrease of plant growth, of chlorophyll, soluble proteins and primary metabolites, but an increase of plant defense markers in the BX WT next maize generation and in plants of the MBOA complementation study. Since the caterpillar growth was reduced they interpreted the results as a suppression of herbivore performance in the next plant generation.

The manuscript contains a lot of work. However, there are several major concerns in the current version regarding manuscript organization, the interpretation of the data, lack of information in the method part and deficiencies in the design of the field experiments.

Major concerns:

- 1) The manuscript is not well organized and difficult to read. It is necessary to switch permanently between the main manuscript and the extended data. The authors should write the manuscript more precisely and important figures should be shown in the main manuscript, for instance Figure 1 can be replaced by Extended Data Figure 1. Concentrations must be included and given in g or mol throughout the entire manuscript.
- 2) There is no information about the history of synthetic herbicide/pesticide use. No determination of possible residual agrochemicals in the soil was done, although their presence could influence the results. The determination seems to be necessary, since the experiments were obviously not done in an organic farming system. There is no proper description of the field culture designs and fertilization regimes.

3) *Spondoptera frugiperda* was described as unaffected by MBOA and to be able to detoxify the compound. The authors know the published data because they cite Maag et al. (2014) in the method paragraph (Plant and insect resources). What is the reason for using an insensitive species? The authors should explain why they chose *S. frugiperda* for the study.

4) An advantage for the next plant generation is not transparent. The plants have a lower protein and chlorophyll content, lower amino acid and sugar contents. The plants are under stress, as indicated by the (small) increase of defense marker gene expressions, which nevertheless did not reduce leaf damage by *S. frugiperda* feeding. Thus, what helps a reduced growth of the caterpillars? Concerning the contents of SA and JA, the known negative crosstalk is not discussed. I wonder about the harvest yields from the next generation when plants are suffering from lower chlorophyll and have to invest energy for long term defense reactions. Regarding these results, benzoxazinoid richness in maize seems to have more negative than positive effects. Phenotyping reveals an herbicidal effect of WT BX73 soils. In agricultural practice crop rotation is used to avoid such effects.

5) The authors found mainly MBOA in the soil cores which were conditioned by wild type (WT) B73, with average concentrations of 2µg/L. Intrinsically, the authors do not know what amounts of MBOA accumulate in the field soil during the vegetation period. No monitoring was performed. The determination of "Average MBOA concentrations in the soils of wild type (WT) B73 plants over 16 weeks of cultivation" was done with pots. The "average concentrations of BXs on the root surface of one week old wild type" is here not relevant, because the microbiomes of old plants were investigated. Since released benzoxazinoids can be degraded by many microorganisms, negative effects on the biomass must have other reasons. The authors argue that all alterations are finally due to MBOA manipulated microbiome compositions, but is there a noticeable response of microorganisms?

6) The demonstrated alteration of root-associated microbial communities is not striking (BX-dependent OTUs are minimal, most OTUs are unchanged), perhaps because fungal and bacterial communities were profiled (or only presented?) with low taxonomic resolution. Moreover the used methods give no robust data about the viability of microorganisms. Are 2µg/L MBOA too low to cause strong alterations in the soil microbiome or were the field cultures just too old and pronounced shifting in the microbial composition is already partly reversed (Wagner et al. 2016: "Bacterial communities changed as host plants aged", article cited by the authors; Johnston-Monje et al. 2016)? The questions cannot be answered, because the authors did not perform microbiome studies after application of different MBOA concentrations to the soil and microbiome analyses were not done with young cultures. The many beneficial microorganisms found by other researchers associated with young maize roots or in the rhizosphere seem not to exist in the samples investigated here. The presence or absence of pathogens is even more important, but no information is given. The determination of the soil microbiome composition before starting the experiment is missing, soil microbiomes are not compared to plant microbiomes, which would be a good control to estimate MBOA induced alterations in the dynamics of species composition and diversity. Culturable microorganisms isolated from the microbiomes were not identified and tested for their responses to MBOA. Although the majority of microorganism are certainly uncultivable, such experiments would give information about microbial properties that help them to cope with the compound and to estimate conversion capacities such as the production of phenoxazinones and their degradation.

7) It is unclear whether and which other compounds are exuded by the roots, since only BXs and maysin were analyzed. It is known that, for instance, organic acids or simple phenylpropanoids can change the microbial species composition. Simple phenolics including salicylic acid have also negative impacts on *S. frugiperda* and plant growth. Microorganisms are known to increase the content of such compounds.

According to my opinion, the statement "Root exudates (MBOA) determine plant growth and defense by shaping the rhizosphere microbiota" stays hypothetical, the results are descriptive. The core question how microorganisms induce impacts on the next plant generation when they are exposed to MBOA is not answered. Another set of experiments seems to be necessary to gain more clarity, such

as:

1. Comparison of the here described microbiomes with ones obtained from young plants with higher taxonomic resolution.
2. Isolation of culturable microorganisms and their treatment with root exudates from young and old plants and with MBOA.
3. Incubation of maize with the isolated microorganisms and feeding experiments with caterpillars from sensitive and insensitive species.
4. Identification of additional simple phenolics in root exudates and leaves.

Perhaps the new results will strengthen the idea that benzoxazinoids have an influence of the subsequent generation via microbiomes.

Point-by-point response letter

Here we respond to the reviewers' comments on the manuscript entitled 'Root exudates determine plant growth and defense by shaping the rhizosphere microbiota' by Hu *et al.* (NCOMMS-17-25861-T). We are grateful for the critical assessment of our work by the referees, as their constructive feedback has substantially contributed to the improvement of our manuscript. Below you find our point-by-point responses to each comment made by the reviewers.

Reviewer #1 (Remarks to the Author):

Hu et al describe a series of experiments that show quite convincingly that the root exudate MBOA, which accumulates in the soil after maize is grown there, causes a shift in the soil microbiota that in turn impacts the growth and level of defense of the next generation of maize. This understanding is achieved with a combination of plant mutants that do not produce benzoxazinoids (which breaks down to produce MBOA), soil sterilization experiments to isolate the critical role of the microbiota, MBOA addition to isolate the importance of that particular exudate, following experiments to assess agricultural importance, microbiota characterization to show plausibility and experiments using a second plant background. I believe that they have made a solid case for a novel finding. The paper is well written, concise and clear, and a pleasure to read.

There were a few things that I would have liked to see considered, although none of them rise to the importance of impacting my overall positive impression of the work.

1. There is an argument about extended phenotypes in the summary. We do know, however, that there can be substantial variation in the microbiota associated with particular plant genotypes. The most comprehensive examination was in Nature Communications by Horton et al. (2014: 5:5320 doi: 10.1038/ncomms6320) and is curiously not cited in this paper.

We thank the reviewer for pointing out this important paper on phyllosphere microbiota. It is now cited in the final paragraph of the manuscript (line 267).

It seems that the impact on the microbiota achieved by the W22 genotype impacts the B73 genotype (although not its own biomass). This doesn't seem consistent with the idea of an extended phenotype. How does heritable variation in the microbiome impact this interpretation of your results? What might be the importance of individual differences in the microbiome?

The referee picked up an important point. While the concept of extended phenotype s applies well to the situation where the host and its microbiota interact in real time (which would be the case for the first generation of maize in our study), it is more difficult to apply it to feedback effects (effects on the second generation of maize in our study). Therefore, we have removed this argument from the summary.

As the reviewer points out correctly, the reduction in plant biomass following BX soil conditioning seems to depend on the plant genotype, while the induction of defenses seems to be conserved. Thus, heritable variation seems to play a role in a subset of BX-dependent plant-soil feedback effects. We now discuss this phenomenon along with potential underlying mechanisms in more detail in the revised manuscript (lines 225-249, see also response below).

2. It is observed that W22 does not share a benefit of modifying its microbiome as was shown for B73. What might drive this difference? It would be interesting to map host variation in this ability!

We would like to rectify the comment that 'W22 does not share a benefit similar to B73'. Our data shows that BX-soil conditioning renders the responding plants more resistant to herbivory. This feedback effect can be considered as a benefit and is shared between W22 and B73.

When it comes to the growth effects (B73 responds with a growth penalty on BX soil conditioning, while this is not seen in the W22 background), we agree that the differential responsiveness in growth is a very interesting finding. B73 and W22 show pronounced genetic differences. A recent paper using single-molecular real-time sequencing and high-resolution optical mapping for instance only mapped 39% of a newly generated W22 map to the B73 genome (Jiao *et al.*, 2017). Therefore, numerous mechanisms may be responsible for the absence of growth reduction in W22 plants growing in W22 BX+ conditioned soils, including altered interactions with root-associated microbes, differential susceptibility to microbe-derived signaling molecules or differences in cross-talk between growth and defense pathways. We agree that mapping this variation is an exciting prospect.

To emphasize the genotype-specific effects found in this study, we now discuss differences in responsiveness of the different genotypes along with possible mechanisms in the revised manuscript (lines 225-249). Therefore, we moved the panels a) and b) of the previous Supplementary Figure 3 to a main Figure (Now Figure 3). The previous panels c) and d) were removed, as they came from an experiment that was not full factorial and was conducted under slightly different environmental conditions. In addition, we now present the new data of the BX-dependent primary metabolite and defense phenotypes of W22 and B73 plants in panel c) of Figure 3 (see also response to Reviewer #2).

3. The importance of the increases in SA and JA seem to warrant some discussion. First, since there is well known crosstalk then why are both increasing? Second, do I get it right that these hormones increase in concentration with no impact on downstream triggering of the pathway? If so, why does resistance to herbivory increase?

An increase in both SA and JA levels in response to stress is not uncommon in plants. Herbivore-attack in *Nicotiana attenuata* for instance increases the biosynthesis of both hormones (Wu *et al.*, 2007), and transcriptional responses of maize to herbivory include both SA and JA marker genes. Negative cross-talk between SA and JA has been established in a number of plant species. In maize however, SA treatment has been found to prime JA production and downstream defenses (Engelberth *et al.*, 2011).

In our experiments, we found that BX conditioning increased the concentration JA marker genes such as the serine protease *ZmSerPIN* and the well-known resistance factor *ZmRIP2*. Thus, it is no surprise that herbivore resistance increases in plants growing in BX conditioned soils. Interestingly, the SA marker genes *ZmPR1* and *ZmPR5* did not respond to BX-soil conditioning, indicating that the SA pathway was not activated despite an increase in SA (note that *ZmPR10*, which was induced by BX-soil conditioning, is JA inducible in maize). The absence of *ZmPR1* and *ZmPR5* induction may thus be

the result of negative cross-talk between JA and SA. These aspects are now discussed in the manuscript (lines 202-224).

4. Are the compositional changes in OTU relative abundances consistent across the two plant host genotypes?

The reviewer asks a highly interesting question; however, we regret that we cannot answer at the moment, as we have not yet investigated the root and rhizosphere microbiomes of W22 plants. We have initiated a PhD study to investigate the root and rhizosphere microbiota of different maize genotypes, including W22. See also our reply to Reviewer #2's comment on the 'broader relevance' of BX-dependent PSFs.

Reviewer #2 (Remarks to the Author):

Hu et al demonstrate 1 week old bx1 seedlings roots produce less BXs compared to wt plants. They also demonstrate that field soil contains detectable levels of BX1s by 3 month old wt plants, while soil from 3 month old bx1 plant BXs are not detectable. They hypothesize the BXs in the soil may influence the microbiome, which is consistent with previous studies demonstrating BX can have negative and positive impacts on microbes. Using the same site they collect root, rhizosphere and bulk soil (3-month-old wt and bx1 plants) and examine bacterial and fungal communities using 16s and ITS sequencing. They demonstrate the soil, rhizosphere, and root microbiota are clearly differed from each other, the differences between wt and bx1 samples were not as clear.

The authors hypothesized that BXs induced changes in soil may influence the next crop planted in the soil. To test this soil from the same site was taken to the lab/greenhouse and wt plants were grown in both types (bx1 conditioned soil and wt conditioned soil). They observed plants were bigger (Dry Weight- Was water content different), had more nutrients, and fewer defenses in leaf tissue when wt plants were grown in soil from bx1 plots compared wt plots. This translated to more palatable plants for herbivores as evidenced in bigger caterpillars on plants grown in bx1 soil compared to wt soil. Taken together these results suggest bx1 plants influenced component of the soil at this site that are important for plant health and defense.

The authors next try to demonstrate which BXs is driving changes in soil by performing a complementation experiment, (although they only examine the impact of 1 BX that was changing in the soil). They take soil from the same field the next year and condition it in the lab with wt and bx1 plants and then grow wt plants in the soil from both treatments. In half the plants growing in bx1 soil they add the BX MBOA. This 'rescues' the wild type soil phenotype (reduced insect growth, reduced plant growth, increased defenses, decreased nutrients) suggesting MBOA is the BX driving changes among treatments. Finally, they try to pin down the component of soil health the BXs are influencing at this site. Using chemical analysis and a series of subtractive experiments (homogenizing soil and sterilizing soil) they demonstrate microbes are important for changes in plant in bx1 soils.

Nice paper and a hard work. There are many studies demonstrating BXs can have negative and positive impacts on microbes and on the relationship of BXs in induction of plant defense. This study does a nice job taking it one step further demonstrating at this site BX1 induced changes in microbes-plant interactions and this influenced plant defense potential to above ground herbivores.

Questions and concerns:

I wonder how the results would differ if soil was taken from another site with a different microbiota? Or if potting soil (without field soil) was used in the lab and conditioned with wt and bx1 plants before being used in the experiments? I expect the impact on plant defense in the next generation largely depends on what microbes are present.

The reviewer points to the important question of the broader relevance of BX-dependent plant-soil feedbacks. We have found functional BX-dependent PSFs (BXs reducing plant growth and increasing resistance to herbivory) in potting soil [**Editorial Note: Reference to unpublished data redacted**]. We furthermore initiated a PhD study on this question where we investigate 2 additional soil types for BX-dependent PSFs. In addition to the PSF phenotypes, we are characterizing the physiological responses of the plant as well as profiling the root and rhizosphere microbes. We will report this study as a new manuscript focusing only on this specific question. We amended the discussion section that future experiments using different soil types are needed to clarify the broader relevance of BX-dependent PSFs (lines 239-249).

[Editorial Note: Unpublished data redacted from Peer Review File as per authorial request.]

Maize is typically rotated as was done at the authors field site (6-year rotation stated by authors). It is not clear if BXs would still be in the soil after crop rotation or if changes in the soil microbiome would still be evident after rotation and able to influence the next generation of maize? It seems the next cover crop or the cover crop immediately prior to planting maize again will likely have a larger influence on the microbiome, compared to BX mediated changes in microbes several years before.

It is important to point out that maize is often planted in monocultures (intensive agriculture in the US, Brazil, France, Germany...) and even in Switzerland with mandatory rotation in conventional farming, maize is sometimes planted into the same field during two consecutive years.

However, the reviewer is right that crop rotation is widespread and generally recommended as more environmentally friendly and sustainable. In this context, the reviewer asks an important question related to the persistence of BXs in soils and the role of BXs in crop rotations. Whether BX soil conditioning would affect other crop plants and whether the induced changes would be strong enough to affect maize performance after several years in a crop rotation remains to be investigated. We now point out in the discussion section that these questions need to be addressed in future studies (lines 225-249).

Could using another plant species or genotype that also alters the soil microbiome result in decreased plant defense in the next generation? Maybe it is not specific to the BXs, just changes in microbiome? The complementation experiment suggests specificity but discussing the previous studies would clarify how this study is different. Previous studies have shown crop rotation can alter nutrients and increase disease incidence and plant susceptibility in the following generation. Changes in/influence of microbiome have also been measured in some of these systems.

We have integrated these suggestions into the introduction and discussion of the manuscript. It is well-known that different plant species have species-specific rhizosphere microbiomes (Bulgarelli *et al.*, 2013) and also the general concept that variation in soil microbiota can be responsible for feedback effects is well established (Bever *et al.*, 2012). Consequently, it is also well-known that different plant species have different capacities to induce feedback effects and that feedbacks can include a decrease in defenses (Klironomos, 2002; van der Putten *et al.*, 2013). The novelty of our study is that we provide with the BX root exudates a mechanistic link between different plant genotypes, their differing microbial communities and their differential feedbacks. Consistently, we find that the BX mutants, exhibiting microbiomes that differ compared from WT, reveal decreased defenses in the next plant generation.

With regard to the 'nutrient' comment: The soil nutrient analysis (Supplementary Table S2) did not reveal differences between BX+ and BX- soils and in addition, we conducted the maize growth experiments at relatively high fertilization levels (see Materials and methods, lines 279-285) to exclude that nutrient aspects confound our findings. Hence, we think that the observed feedback effects are independent of soil nutrient status.

With regard to the 'different plant species' comment: In our efforts to clarify the broader relevance of BX-dependent plant-soil feedbacks (PhD project, see above), we started to examine the effects of BX exudation on wheat as a crop that can be planted after maize. We mention now in the discussion section that future experiments using different plant species are needed to clarify the broader relevance of BX-dependent PSFs in crop-rotation systems (lines 240-249).

Why are BX concentration presented so different from figure to figure? (Expressed relative to cm² of root, per L, per pot). It is hard to compare across experiments the relative concentrations observed and used. Can you standardize the presentation so it is easier to compare across experiments? Maybe this is not possible.

The units are different due to the different experimental protocols and sampling units (soil volumes vs. root systems). Using the same units for all concentrations is, therefore not practical. It would for instance make little sense to express exudation as a function of root mass, as the root surface rather than the mass of the roots determines the exuded amounts. Also, the MBOA kinetics are best expressed as total amounts per plant, as these were the amounts that were used for MBOA complementation.

The bx1 conditioned field soil data suggests no BXs were present at the site from previous maize propagation – so MBOA is not stable for that long? What about the presence of BXs in the conditioned soil before and after each experiment? This would strengthen the connection and potential for impact on the next generation. Is always MBOA the highest BX in the soil? Other BXs were changing significantly in the conditioned field soil (for example - DIMBOA, HDMBOA-glc, HBOA-glc). Could these also be driving changes?

The field was planted with other crops for several years before these experiments. MBOA has a half-life of 5-6 days (Etzerodt *et al.*, 2008). It is therefore highly unlikely that MBOA from previous maize propagation was present in the field. MBOA catabolites such as AMPO are more stable and may

persist for years in the soil. However, we detected only very low concentrations of AMPO in our soil samples and did not find any differences in concentration of this metabolite between WT and *bx1* cultivated soils. Thus, differences in AMPO levels cannot explain the observed feedback effects.

The reviewer is right that MBOA is not the only benzoxazinoid, which differs in the soils of WT and *bx1* plants. Depending on the time of sampling, other benzoxazinoids may also be present in high concentrations and may affect soil microbiota and feedback effects. Our MBOA complementation experiments demonstrate however that MBOA is both necessary and sufficient to trigger the observed feedback effects. We discuss our results accordingly (lines 194-200).

The W22 genotype experiment was nice – but plant defenses and nutrients were not measured in the plants growing in conditioned soils. Also there was no impact on plant growth for w22 experiment, although insect growth was still increased in the *bx1/w22*. Is this also due to a lack of defense or could it be due to only increased nutrient availability? This experiment should be done if it was not already.

We agree with the reviewer and performed this missing experiment (lines 121-125). The new data was added to the manuscript as the new Fig. 3c and Supplementary Fig. 3. We find similar defense responses in B73 and W22 maize backgrounds that are consistent with their feedback phenotypes. Interestingly, leaf soluble protein contents were not changed in W22, which is consistent with the absence of growth phenotypes and suggests that the suppression of *S. frugiperda* growth is due to changes in the plant's defense status rather than reduced nutrient availability. This is now discussed in the manuscript (lines 225-238).

It would have been nice if the authors also conditioned potting soil or some other soil with *bx1* and *wt* and demonstrated the same findings. It may be very specific finding to this one site and set of microbes and the authors should address this with experiments or in the discussion.

We refer to our detailed reply to the first question/concern of Referee #2.

We find similar effects in another soil type [**Editorial Note: Reference to unpublished data redacted**]. However, this dataset provides little added value on its own and needs to be combined with a broader assessment of different field soils along with the corresponding microbiome changes to be informative regarding the role of different initial microbial communities in BX-dependent feedback effects. Performing all these experiments are out of the scope of the current manuscript, but will be conducted over the next year in the context of a follow-up manuscript. However, as suggested by the reviewer, we now discuss this aspect in the revised version of the manuscript (lines 240-249).

Before the lab experiments the field 'conditioned' soil was analyzed for changes in chemistry/health. No changes in soil chemistry/health were observed in the field collected samples. Could soil chemistry/health be changing over the 3 months of the greenhouse/lab experiments? The microbe populations could still be driving the changes. This would tell you if changes in soil chemistry from the microbes or direct plant-microbe interactions are important for the phenotype in the next generation.

The reviewer makes an excellent suggestion. We have analyzed the soil chemical parameters in soil cores collected at the end of a feedback experiment. We do not find any significant difference in soil chemistry between BX+ and BX- soil cores after the feedback growth period of 3 months. The new data was added as Supplementary Table 3 to the results section of the manuscript (lines 147-148).

Legend missing on extended figure 1 16s and ITS data.

We are grateful for spotting this mistake and have complemented the manuscript with the missing legend.

Reviewer #3 (Remarks to the Author):

As the major claim, the manuscript deals with the question whether benzoxazinoids can function as modifiers of plant microbiome composition resulting in impacts on the next plant generation. Regarding benzoxazinoids comparable studies have not been done, but effects of other secondary metabolites on subsequent cultures are known. The idea that plant secondary metabolite triggered microbiomes induce systemic resistance against pathogens and herbivores is not new. Such influences are important for agriculture and they are therefore of general interest.

In detail, the authors investigated possible influences of MBOA on the soil and root microbiota, subsequent impacts on maize and growth of *Spodoptera frugiperda* caterpillars feeding on leaves of these maize plants. They used BX 73 or mutants as soil conditioner in comparison to non-conditioned field soils and performed complementation experiments with MBOA. They found a decrease of plant growth, of chlorophyll, soluble proteins and primary metabolites, but an increase of plant defense markers in the BX WT next maize generation and in plants of the MBOA complementation study. Since the caterpillar growth was reduced they interpreted the results as a suppression of herbivore performance in the next plant generation.

The starting point of the study was not to investigate possible influences of MBOA. We studied the hypothesis that the secretion of BX impacts the root microbiota. Having found that BXs affected the rhizosphere and root microbes, we tested if the changes in microbial communities affect plant and herbivore growth. The results revealed that the differential microbial communities provoke differing plant phenotypes. We then found that MBOA accumulates in the soil and is both necessary and sufficient to explain these microbiome-dependent effects. Importantly, we do not compare B73- ("BX 73" plants do not exist) and *bx1*-conditioned soils to non-conditioned soils. Instead, and as the other reviewers understood correctly, we specifically compared relative feedback effects of wild type and BX-deficient maize plants.

We improved the clarity of our manuscript by 1) stressing in the title that root exudate metabolites drive plant-soil feedbacks by shaping the rhizosphere microbiome and 2) by introducing a uniform vocabulary to avoid confusion between plant genotypes and soil conditioning. We now term the differentially conditioned soil cores as 'BX+' and 'BX-' for WT- and mutant-conditioned soils, respectively (lines 96-98). Hence, BX+ soils were conditioned by BX secreting B73 plants, and BX- soils were conditioned by BX defective *bx1* plants.

The manuscript contains a lot of work. However, there are several major concerns in the current version regarding manuscript organization, the interpretation of the data, lack of information in the method part and deficiencies in the design of the field experiments.

Major concerns:

1) The manuscript is not well organized and difficult to read. It is necessary to switch permanently between the main manuscript and the extended data. The authors should write the manuscript more precisely and important figures should be shown in the main manuscript, for instance Figure 1 can be replaced by Extended Data Figure 1.

We apologize for the difficulties the reviewer may have had in reading the manuscript. The manuscript was transferred from *Nature*, and the format for *Nature* articles restricted our capacity to include major results into the main figures. We have now reformatted the article for *Nature Communications*, which allowed us to reorganize the figures. For Figure 1, it is not practical to show both the graphs from the main and extended Figure together. Whether to show chromatograms or bar graphs and whether to show CAPs or unconstrained ordinations of different microbiota is a matter of personal preference. From an analytical point of view, we think that the chromatograms are more informative than the bargraphs and include the important information regarding the differences in BX exudation and accumulation between WT and *bx1* mutant plants. A similar argument applies to the microbiota profiles, where the ordinations are suitable to transmit the message (microbiome variation as a function of BX exudation). We now reorganized the manuscript to be accessible to a broad readership. The main figures report the main conclusions, while the supplements provide more details for specialists in the field.

Concentrations must be included and given in g or mol throughout the entire manuscript.

All BX concentrations are given in μg throughout the manuscript. We do not understand why these should be changed into g. See also our response to Reviewer #2 regarding the different units of measurement and why they were chosen.

2) There is no information about the history of synthetic herbicide/pesticide use. No determination of possible residual agrochemicals in the soil was done, although their presence could influence the results. The determination seems to be necessary, since the experiments were obviously not done in an organic farming system. There is no proper description of the field culture designs and fertilization regimes.

We describe the cultivation history of the field in the manuscript (lines 279-285). The field was managed according to conventional farming practices ('conventional' implies the use agrochemicals such as pesticides and mineral fertilizers; this is made more explicit in the revised version). The entire field was subjected to the same practices and all experiments were performed with this one field soil. Thus, possible residual agrochemicals do not confound our results. We agree that understanding how different cropping practices affect microbiota-mediated feedback effects would be interesting. However, this question goes beyond the scope of the presented work.

3) *Spondoptera frugiperda* was described as unaffected by MBOA and to be able to detoxify the compound. The authors know the published data because they cite Maag et al. (2014) in the method paragraph (Plant and insect resources). What is the reason for using an insensitive species? The authors should explain why they chose *S. frugiperda* for the study.

We have difficulties to understand this comment. We have never tested for direct effects of BXs, including MBOA, on herbivores, but were interested in how BX release into the soil affects the rhizosphere microbes and how these microbial communities impact the resistance patterns of the next plant generation. Furthermore, we document that the differential microbial communities do not affect BX levels in the leaves of next plant generation.

Spodoptera frugiperda was chosen because it is an important and invasive leaf pest of maize. This is now described in the manuscript. *S. frugiperda* is tolerant to MBOA, but susceptible to other benzoxazinoids (Glaser *et al.*, 2011) and maize defenses such as RIP2 (Chuang *et al.*, 2014).

4) An advantage for the next plant generation is not transparent. The plants have a lower protein and chlorophyll content, lower amino acid and sugar contents. The plants are under stress, as indicated by the (small) increase of defense marker gene expressions, which nevertheless did not reduce leaf damage by *S. frugiperda* feeding. Thus, what helps a reduced growth of the caterpillars? Concerning the contents of SA and JA, the known negative crosstalk is not discussed. I wonder about the harvest yields from the next generation when plants are suffering from lower chlorophyll and have to invest energy for long term defense reactions. Regarding these results, benzoxazinoid richness in maize seems to have more negative than positive effects. Phenotyping reveals an herbicidal effect of WT BX73 soils. In agricultural practice crop rotation is used to avoid such effects.

We identify numerous critique points and reply to them individually:

BX exudation causes enhanced defenses (the reverse conclusion of 'reduced caterpillar growth'; can be seen as advantage) but at the same time penalizes on plant growth in B73 background (a disadvantage). Therefore, we carefully report the feedback results in a neutral way without suggesting an "advantage" for the next plant generation.

SA and JA: We agree with the reviewer that cross-talk between SA and JA warrants a careful discussion and have added a whole paragraph on this issue to the manuscript (lines 202-224). See also our reply Reviewer #1.

Yield/rotation: The reviewer raises an important point. As pointed out in our responses to reviewer #2, maize can and is often planted in monoculture in intensive farming systems. We initiated a PhD study on the broader relevance of BX-dependent plant-soil feedbacks and this project includes a field experiment, where the yield and rotation questions will be assessed. This however, presents a 2-year effort that is beyond the scope of this study. The purpose of our study was not to examine the agricultural relevance of these effects, but we mention potential inference for agriculture in the discussion section (lines 240-249).

5) The authors found mainly MBOA in the soil cores which were conditioned by wild type (WT) B73,

with average concentrations of 2µg/L. Intrinsically, the authors do not know what amounts of MBOA accumulate in the field soil during the vegetation period. No monitoring was performed. The determination of “Average MBOA concentrations in the soils of wild type (WT) B73 plants over 16 weeks of cultivation” was done with pots. The “average concentrations of BXs on the root surface of one week old wild type” is here not relevant, because the microbiomes of old plants were investigated. Since released benzoxazinoids can be degraded by many microorganisms, negative effects on the biomass must have other reasons. The authors argue that all alterations are finally due to MBOA manipulated microbiome compositions, but is there a noticeable response of microorganisms?

We think the reviewer confounds the results of two distinct experiments.

(1) We measured the BX levels in the soil cores at the end of the cultivation period in the field, because we wanted to understand how much BX was present at the beginning of the feedback experiment with the next plant generation (here we found that MBOA was the accumulating form). We disagree that a monitoring of MBOA in the field prior to subsequent soil-feedback effects would be helpful.

(2) The second experiment, where we determined MBOA concentrations (weekly time-course measures in the pot experiment), was conducted to determine precisely how much MBOA we need to add to pots each week for the complementation experiment.

With regard to the question, whether microorganisms are involved, the reviewer must have overlooked the sterilization experiment of *bx1* plants that were complemented with MBOA. Because this experiment should not be overlooked, we now present this data in the main text (Fig. 6). This experiment revealed that removal of the microbial communities (sterilization) eliminated all of the effects induced by application of MBOA without changing residual MBOA levels, thus demonstrating that MBOA acts by altering the soil microbiota.

6) The demonstrated alteration of root-associated microbial communities is not striking (BX-dependent OTUs are minimal, most OTUs are unchanged), perhaps because fungal and bacterial communities were profiled (or only presented?) with low taxonomic resolution. Moreover the used methods give no robust data about the viability of microorganisms. Are 2µg/L MBOA too low to cause strong alterations in the soil microbiome or were the field cultures just too old and pronounced shifting in the microbial composition is already partly reversed (Wagner et al. 2016: “Bacterial communities changed as host plants aged”, article cited by the authors; Johnston-Monje et al. 2016)? The questions cannot be answered, because the authors did not perform microbiome studies after application of different MBOA concentrations to the soil and microbiome analyses were not done with young cultures. The many beneficial microorganisms found by other researchers associated with young maize roots or in the rhizosphere seem not to exist in the samples investigated here. The presence or absence of pathogens is even more important, but no information is given. The determination of the soil microbiome composition before starting the experiment is missing, soil microbiomes are not compared to plant microbiomes, which would be a good control to estimate MBOA induced alterations in the dynamics of species composition and diversity. Culturable microorganisms isolated from the microbiomes were not identified and tested for their responses to MBOA. Although the majority of microorganism are certainly uncultivable, such experiments would

give information about microbial properties that help them to cope with the compound and to estimate conversion capacities such as the production of phenoxazinones and their degradation.

This paragraph contains many different discussion points. We tried to disentangle the different statements and reply to them individually.

- We strongly disagree with the reviewer's interpretation about the effect size of BX-dependent alterations of the root microbiota (~5% of total microbiota variation). We have stated in the manuscript that this level of variation is commonly seen between different accessions or varieties (see Hacquard *et al.*, 2015). While these are multi-loci comparisons, we compare NILs which differ at one locus and find the same degree of microbiota variation as others have found when comparing 27 different maize inbred lines (Peiffer *et al.*, 2013). We have added a whole paragraph on this issue to the discussion section (lines 240-249).

- We utilized state-of-the-art microbiota profiling commonly applied by leading laboratories around the globe (16S: Schulze-Lefert at MPI Cologne; ITS: Fierer at University of Colorado). While we agree on inherent limitations of such approaches (low taxonomic resolution and non-discrimination of 'relic/live DNAs'), this criticism is unnecessary here, as the limitations of the method do not confound the interpretation of our data. We see for instance robust differences between genotypes and between differentially conditioned soils. With regard to viability of the microbes, the key results of our study are that the removal of preconditioned microbial communities by X-ray abolished the observed plant responses and that the reintroduction of microbes (extracts of the differentially preconditioned communities) restored the differential phenotypes.

- The reviewer raises the concern that MBOA levels are too low or the soil cores could be too old to provoke pronounced microbiota shifts. We agree there might be temporal BX-dependent microbiota dynamics (possibly, microbiota differences are more pronounced at earlier sampling time points), but it is a moot point to dispute about required effect sizes of microbiome shifts that would be necessary to produce the differential plant phenotypes. Facts are that with starting pot cultures differing at MBOA levels of 2µl/L, we see differences in microbiota composition at 3 months, the sterilization/re-inoculation experiments demonstrated the involvement of the microbiota and we robustly measure the differential plant phenotypes (also at 3 months). Of note, the legacy effect is long lasting; we find the differential plant phenotypes also examining soil cores that we left overwinter for a long time in the field (Supplementary Fig. 4). The work by Wagner *et al.* (2016) is not useful for the assumption that the microbial communities have reversed. They examined perennial *Boechera* and found age effects over a period of 2 to 4 years, while we examined microbial communities of annual maize after a short cultivation of 3 months. Moreover, host genetic control of root bacterial communities was not demonstrated in the Wagner paper.

- Performing microbiome response studies after application of different MBOA concentrations to soil is a nice suggestion for the future, but goes beyond the scope of this study. Here we report the phenomenon of the BX-dependent growth phenotypes together with the underlying mechanism.

- We are criticized not to include information about beneficial or pathogenic microbiome members. We decided not to discuss or speculate about different types of interactions, because the short OTU-sequence based taxonomy is insufficient to provide reliable functional information. Bacteria with the exact same 16S rRNA gene sequence can be beneficial or pathogenic (e.g. *Pseudomonas*). We are of

the opinion that functional interpretations should be reserved to metagenome (potential functions), transcript or protein analyses.

- Instead of sampling soil before starting the experiment, we compared samples from soil cores (collected at the edge of the soil cores, distantly from roots), which were conditioned by either B73 or bx1 plants during the growth period of 3 months. We think this is more meaningful, because these soil samples are comparable to the root and rhizosphere samples as they have the same cultivation history. Therefore, we think that the suggested experiments will not yield meaningful new insights for this study.

- The comment that microorganisms were not isolated and characterized for their responses to MBOA is asking for an effort that is well beyond the scope of this work. Our future goal is to identify the microbiome players that are responsible for the observed feedback effects, and for this we will work with isolated rhizobacteria. We agree with the reviewer that we would learn from such experiments about the microbial traits required for responding and tolerating BX compounds, however, this opens a completely new line of research.

7) It is unclear whether and which other compounds are exuded by the roots, since only BXs and maysin were analyzed. It is known that, for instance, organic acids or simple phenylpropanoids can change the microbial species composition. Simple phenolics including salicylic acid have also negative impacts on *S. frugiperda* and plant growth. Microorganisms are known to increase the content of such compounds.

We did not analyze maysin exudation, but determined the levels of this metabolite in the leaves, as it is a known resistance factor against *S. frugiperda*. Maysin has never been documented to be released by maize roots. Our goal was never a holistic examination of maize root exudates but we focused on the targeted investigation of BXs in root exudates (therefore we use WT and mutant plants in the BX pathway). With regard to the comment of SA and its negative effects on *S. frugiperda* and plant growth: we indeed show increased leaf SA levels and reduced *S. frugiperda* growth and plant growth.

According to my opinion, the statement "Root exudates (MBOA) determine plant growth and defense by shaping the rhizosphere microbiota" stays hypothetical, the results are descriptive. The core question how microorganisms induce impacts on the next plant generation when they are exposed to MBOA is not answered. Another set of experiments seems to be necessary to gain more clarity, such as:

1. Comparison of the here described microbiomes with ones obtained from young plants with higher taxonomic resolution.
2. Isolation of culturable microorganisms and their treatment with root exudates from young and old plants and with MBOA.
3. Incubation of maize with the isolated microorganisms and feeding experiments with caterpillars from sensitive and insensitive species.
4. Identification of additional simple phenolics in root exudates and leaves. Perhaps the new results will strengthen the idea that benzoxazinoids have an influence of the subsequent generation via microbiomes.

We have responded in detail to the requested set of experiments in our responses above. We agree that the proposed experiments would give additional insights into our work. However, the requested experiments are beyond reasonable scale and we consider them as suggestions for future scientific endeavor. Importantly, the requested experiments do not challenge the presented findings, but open new lines of research and tackle additional research questions. Although, we have not yet identified the microbe(s) that actively translate(s) the root exudation signal(s) of the plant into the growth and resistance phenotypes of the subsequent plant generation, we would like to point out that our study significantly advances the field and pushes the boundaries of knowledge substantially: our work reveals (1) that the exudation of BXs alters the root and rhizosphere communities and (2) we show that this affects growth and resistance traits of a subsequent plant generation growing in BX conditioned soils. Moreover, (3) by using sterilization and microbial complementation treatments we demonstrate the involvement of the microbiota and (4) with chemical complementation we identify MBOA as driving substance for the observed plant phenotypes. We are unaware of any other study to document plant-soil feedback effects at this level of mechanistic detail.

References

- Bever JD, Platt TG, Morton ER. 2012.** Microbial population and community dynamics on plant roots and their feedbacks on plant communities. *Annual Review of Microbiology* **66**: 265-283.
- Bulgarelli D, Schlaeppi K, Spaepen S, Ver Loren van Themaat E, Schulze-Lefert P. 2013.** Structure and functions of the bacterial microbiota of plants. *Annual Review of Plant Biology* **64**: 807-838.
- Chuang WP, Herde M, Ray S, Castano-Duque L, Howe GA, Luthe DS. 2014.** Caterpillar attack triggers accumulation of the toxic maize protein RIP2. *New Phytologist* **201**: 928-939.
- Engelberth J, Viswanathan S, Engelberth MJ. 2011.** Low concentrations of salicylic acid stimulate insect elicitor responses in *Zea mays* seedlings. *Journal of Chemical Ecology* **37**: 263-266.
- Etzerodt T, Mortensen AG, Fomsgaard IS. 2008.** Transformation kinetics of 6-methoxybenzoxazolin-2-one in soil. *Journal of Environmental Science and Health Part B-Pesticides Food Contaminants and Agricultural Wastes* **43**: 1-7.
- Glaser G, Marti G, Villard N, Doyen GA, Wolfender JL, Turlings TC, Erb M. 2011.** Induction and detoxification of maize 1,4-benzoxazin-3-ones by insect herbivores. *Plant Journal* **68**: 901-911.
- Hacquard S, Garrido-Oter R, Gonzalez A, Spaepen S, Ackermann G, Lebeis S, McHardy AC, Dangl JL, Knight R, Ley R, et al. 2015.** Microbiota and host nutrition across plant and animal kingdoms. *Cell Host & Microbe* **17**: 603-616.
- Jiao Y, Peluso P, Shi J, Liang T, Stitzer MC, Wang B, Campbell MS, Stein JC, Wei X, Chin C-S. 2017.** Improved maize reference genome with single-molecule technologies. *Nature*.

- Klironomos JN. 2002.** Feedback with soil biota contributes to plant rarity and invasiveness in communities. *Nature* **417**: 67-70.
- Peiffer JA, Spor A, Koren O, Jin Z, Tringe SG, Dangl JL, Buckler ES, Ley RE. 2013.** Diversity and heritability of the maize rhizosphere microbiome under field conditions. *Proceedings of the National Academy of Sciences of the United States of America* **110**: 6548-6553.
- van der Putten WH, Bardgett RD, Bever JD, Bezemer TM, Casper BB, Fukami T, Kardol P, Klironomos JN, Kulmatiski A, Schweitzer JA, et al. 2013.** Plant-soil feedbacks: the past, the present and future challenges. *Journal of Ecology* **101**: 265-276.
- Wu J, Hettenhausen C, Meldau S, Baldwin IT. 2007.** Herbivory rapidly activates MAPK signaling in attacked and unattacked leaf regions but not between leaves of *Nicotiana attenuata*. *Plant Cell* **19**: 1096-1122.

Reviewers' comments:

Reviewer #1 (Remarks to the Author):

Thank you for the care taken in responding to my comments. I am comfortable with the responses and believe that the additions to the manuscript make an already strong piece of work stronger. It was a pleasure to read.

Reviewer #3 (Remarks to the Author):

Comments to the revised Nature Communication manuscript NCOMMS-17-25861A

The authors have responded positively to most of the comments. Also due to the constructive criticisms of the other reviewers, the manuscript is properly revised and improved. It is now easy to read. However, there are still two major concerns which need to be addressed by the authors.

(1). The exclusiveness of MBOA in shaping the composition of rhizosphere microbiota is not convincing. According to Petriacq et al. 2017 (Plant Journal 92, 147–162), the major portion of maize root exuded secondary metabolites are flavonoids (“Strikingly, a relatively large fraction of maize rhizosphere-enriched ions could be annotated to flavonoids (28%) and benzoxazinoids (21%)”). Flavonoids are long known to influence the microbial diversity. Thus the shift in microbial community composition could be due to both classes of compounds at a minimum, acting in concert and resulting in synergistic or additive effects. It would be nice to add an experiment with combinations of the MBOA and maize root exudate flavonoids, in particular as MBOA concentrations are very low. Flavonoids can exist also in sterilized soil. Meanwhile numerous secondary metabolites including the wide spread simple phenolics have been found to modify plant associated microbial communities (for instance Zhou et al. 2018 Biology and Fertility of Soils). Benzoxazinoids may be only additional compounds, among many other secondary metabolites, found to be able to affect microbial compositions.

(2). Criticizing the “state-of-the-art microbiota profiling” is justified. Recently Carini et al. unveil the problem, showing “that extracellular DNA affects molecular analyses of microbial diversity” and that “extracellular DNA inflated the observed prokaryotic and fungal richness by up to 55% and caused significant misestimation of taxon relative abundances, including the relative abundances of taxa integral to key ecosystem processes.”(NATURE MICROBIOLOGY 2, 16242 (2016). This problem should be at least discussed as the authors certainly cannot ascertain their data by other methods that allow insights in the viability of the microorganisms (see perhaps Gruner et al. 2017, Microb. Ecol.).

Reviewer #4 (Remarks to the Author) [Recruited to comment on behalf of Reviewer #2]:

Hu et al., Root exudate metabolites drive plant-soil feedbacks on growth and defense by shaping the rhizosphere microbiota

This is an extremely interesting, if somewhat difficult to follow, manuscript that details interactions between maize plants which secrete benzoxazinoids into soil and the microbiota in the rhizosphere. There are very few studies which demonstrate a causal relationship between root exudate chemistry and rhizosphere communities and so the authors are to be supported in their efforts to do so. Furthermore, since maize is an important agronomic crop, any demonstrated causal relationships could provide a mechanism to improving crop performance and resistance to disease - with great societal benefits, so the work is clearly important. I have no doubt that the work will be of interest to a wide readership of Nature Communications.

The authors have done a thorough job of responding to the comments and criticisms from the first reviews, much of which were concerned with plant-related issues. I do not wish to rehearse these again but instead would like to focus on the analysis of the amplicon sequencing which I feel needs a comprehensive re-evaluation.

To start at the beginning, the authors compare the rhizosphere communities which develop in association with BX+ve and BX-ve maize lines and bulk soil communities by generating bacterial and fungal amplicon datasets. Having established these sequence datasets, they then jump straight in to comparing β -, between sample, diversity using multivariate ordination and permutation multivariate analysis of variance (PERMANOVA).

- I would suggest that it is necessary to compare the α -, within sample, diversity first since the central hypothesis of the study is that BXs alter the root-associated microbial communities. One logical way to examine this is to test whether species richness (or some other measure of α -diversity) is significantly different between BX+ve and BX-ve. This is a trivial test to perform and differences between the measures should be testable using robust parametric statistics such as analysis of variance and post-hoc tests, if there is indeed a significant effect of treatment. The results of this would be informative since one might hypothesize that if BXs act to either attract specific organisms to the rhizosphere or reduce the rhizosphere community due to toxicity effects then the species richness of BX+ve plants may be reduced. My own preference would be to use Chao-1 estimates of richness, but there are plenty of other measures. Why were α -diversity measures not considered?
- The authors next use PERMANOVA to test differences in community structure and show a significant difference between the treatments (bulk soil – WT root – bx1 root – WT rhizosphere – bx1 rhizosphere). There are two issues here: a significant difference observed in PERMANOVA may arise from differences in the dispersion of the data and this can be tested, but wasn't, using the PERMDISP test – I recommend that it be performed here using the "betadisper" code in vegan; secondly, although the PERMANOVA test indicates a significant treatment effect, there are no post-hoc pairwise comparisons of the treatments, the only significant difference could well be the difference between the bulk soil treatment and all the other treatments (it certainly looks possible from the ordinations in Figure 1) - there needs to be some pairwise comparison of the treatment effects.
- Both unconstrained and constrained ordination procedures were used to visualize differences between the communities developed under each treatment. Bray-Curtis distance metrics were used, but there is no justification regarding why. Were other distance metrics used and was the ordination consistent? Why were weighted Unifrac distances not considered, since they are designed for exactly the type of data generated in this study and the paper by Weiss et al. (Microbiome, 2017, 5:27) - which the authors cite - demonstrates that the metric is both accurate and not subject to sequencing depth-related artefacts (and thus not requiring rarefaction of the data). Weighted Unifrac out performs Bray-Curtis consistently.
- Principal Coordinate Analysis (PCoA) is used to demonstrate differences between the treatments and both fungal and bacterial communities. In both cases, well less than 50% of the variation in the experiment was accounted for by the first two principal coordinates, which in my experience in soil microbiology is rather low. For both bacteria and fungi, the main difference seems to be between rhizoplane-, rhizosphere-associated communities and the bulk soil since these differences are ordinated along Axis 1 in both cases. This rhizosphere effect is consistent with a great number of studies and not novel. In the case of bacteria, the rhizoplane community certainly appears to be different from the rhizosphere and bulk soil communities, but the difference is small. Does this analysis improve if weighted Unifrac distances are used?
- Constrained ordination is then used, although exactly why is never explained. Unconstrained ordination attempts to display the variation of multivariate data in a reduced number of dimensions, while constrained ordination attempts to display the variation that can be explained by measured "environmental factors". There is very little explanation of how the constrained ordination was implemented – the authors describe using partial canonical analysis of principal coordinates (CAP), but what factors were used to constrain the analysis? There are a number of factors that could be used,

including the soil compartment (bulk, rhizosphere, rhizoplane), BX concentrations (or presence/absence) since this is the basis for the hypothesis being tested, the data shown in Supplementary Table 2, or a combination of these. Whatever was used appears to do a poor job of explaining the variation, since the amount is less than that explained by the unconstrained ordinations – raising the questions as to whether the constraining factors were chosen correctly. Also, missing from the constrained ordinations is any test of whether the constraining variables account for a significant proportion of the variability. It is therefore impossible to comment on the validity of the models presented. I must admit that I am less familiar with CAP than the more often encountered redundancy analysis or canonical correspondence analysis, but I am not convinced that the CAP shown here is providing any additional insight into the data or perhaps even implemented properly. A redundancy analysis is shown in Figure 5, why not apply it here too?

- The authors describe an interesting complementation experiment where microflora washed from soil conditioned with either BX+ve or BX-ve plants are added back to sterilized soils in an attempt to demonstrate any long-term influence of the established microbiomes on plant metabolism. Redundancy analysis is used here and presents the most compelling evidence in the manuscript for an effect of BXs in the sense that differences between BX+ve and BX-ve soils are apparent along Axis 1 which is associated with 75% of the variability (considerably more than is the case in the unconstrained analysis presented in Supplementary Figure 7) but again, there is no description of which factors were used to constrain the model and no test of whether the model accounts for a significant amount of the total variability. This information must be reported for an informed assessment of the model to be possible.

- Furthermore, regarding the “transplanting” of microbiomes from each soil: what efforts were made to ensure that the comparisons are meaningful? For example, were the numbers of viable cells added to sterilized soils the same, or were extracts just added and assumptions made regarding the number and viability of cells? If the number of viable cells is significantly different between the treatments it is not possible to draw conclusions which relate exclusively to the community composition of the pre-conditioned soils. Nothing fancy here, but a Live-Dead fluorescence assay should help inform this issue. Furthermore, are the extraction procedures properly accounting for any differences in community structure between the different soils? There is no amplicon-based assessment that the differences observed in the soils is replicated by the extracted microflora. The authors make a great many implicit assumptions here for which there is little support.

- In the Methods section (page 26, lines 557-559) the authors describe applying rarefaction to their datasets and reference the Weiss et al. paper. Careful reading of the paper suggests that rarefaction is only necessary if sequence reads differ by more than a factor of 10 between samples. Was this the case? Some detail regarding the depth of sequencing between the different datasets is necessary.

- Why were Cyanobacteria omitted from the analyses? Are they considered not to be part of the soil microbiome?

- Description and justification for the ordination procedures carried out here is necessary. Also, bi-plots of the data and factors used in the constrained ordinations is also necessary, as well as a thorough statistical testing of the models.

Reviewers' comments:

Reviewer #1 (Remarks to the Author):

Thank you for the care taken in responding to my comments. I am comfortable with the responses and believe that the additions to the manuscript make an already strong piece of work stronger. It was a pleasure to read.

Reviewer #3 (Remarks to the Author):

Comments to the revised Nature Communication manuscript NCOMMS-17-25861A

The authors have responded positively to most of the comments. Also due to the constructive criticisms of the other reviewers, the manuscript is properly revised and improved. It is now easy to read. However, there are still two major concerns which need to be addressed by the authors.

(1). The exclusiveness of MBOA in shaping the composition of rhizosphere microbiota is not convincing. According to Petriacq et al. 2017 (Plant Journal 92, 147–162), the major portion of maize root exuded secondary metabolites are flavonoids (“Strikingly, a relatively large fraction of maize rhizosphere-enriched ions could be annotated to flavonoids (28%) and benzoxazinoids (21%)”). Flavonoids are long known to influence the microbial diversity. Thus the shift in microbial community composition could be due to both classes of compounds at a minimum, acting in concert and resulting in synergistic or additive effects. It would be nice to add an experiment with combinations of the MBOA and maize root exudate flavonoids, in particular as MBOA concentrations are very low. Flavonoids can exist also in sterilized soil. Meanwhile numerous secondary metabolites including the wide spread simple phenolics have been found to modify plant associated microbial communities (for instance Zhou et al. 2018 Biology and Fertility of Soils). Benzoxazinoids may be only additional compounds, among many other secondary metabolites, found to be able to affect microbial compositions.

We agree with the reviewer that root exudates contain many other metabolites, which are likely to shape the rhizosphere microbiota. Our work does not claim that benzoxazinoids are exclusive to this process. Rather than that, we took a genetic and pharmacological approach to investigate the contribution of benzoxazinoids to plant-soil feedbacks within a background of other root exudate metabolites. In our opinion, our work convincingly shows that benzoxazinoids have important effects, but this does not by any means exclude effects of other metabolites. Understanding how benzoxazinoids interact with other exudate metabolites is an exciting prospect, but is outside of the scope of the current work. Such experiments would take several years to conduct, especially because the identity and prevalence of other metabolites such as flavonoids in the exudates is not well resolved, and our capacity to control flavonoid exudation is currently very limited. For instance, there are currently no maize double mutants available that are silenced in the production of benzoxazinoids and flavonoids. The following parts of the manuscript deal with this aspect

and clarify that MBOA is not exclusive in determining the composition of the rhizosphere microbiota:

Line 44- beyond: “Plants can change the soil microbiota by secreting bioactive molecules into the rhizosphere. Root exudates typically comprise primary metabolites such as sugars, amino acids, and carboxylic acids as well as a diverse set of secondary metabolites. Besides representing carbon and nitrogen substrates for microbial growth, root exudate compounds have a multitude of effects on rhizosphere microbes by acting as signaling molecules, attractants, stimulants, but also as inhibitors or repellents. Thereby, the composition of root exudates, which is under host-genetic control, likely defines the assembly of plant-specific root and rhizosphere microbial communities.”

Line 56: “Maize root exudates are comprised of a variety of metabolites, including substantial amounts of benzoxazinoids (BXs) that are secreted to the rhizosphere.”

Line 197: “Our findings support the following mechanistic model (Fig. 8): Maize plants release a blend of metabolites, including BXs such as DIMBOA, from the roots and thereby influence the composition of the root-associated microbiota (Fig. 1).”

Line 259: “It will also be interesting to investigate if and how BXs interact with other bioactive exudate metabolites from maize such as flavonoids, which may also shape the maize rhizosphere microbiota.”

(2). Criticizing the “state-of-the-art microbiota profiling” is justified. Recently Carini et al. unveil the problem, showing “that extracellular DNA affects molecular analyses of microbial diversity” and that “extracellular DNA inflated the observed prokaryotic and fungal richness by up to 55% and caused significant misestimation of taxon relative abundances, including the relative abundances of taxa integral to key ecosystem processes.”(NATURE MICROBIOLOGY 2, 16242 (2016). This problem should be at least discussed as the authors certainly cannot ascertain their data by other methods that allow insights in the viability of the microorganisms (see perhaps Gruner et al. 2017, Microb. Ecol.).

We agree with the reviewer’s concern that abundant relic DNA in soil obscures estimates of soil microbial diversity. While relic DNA presents a serious problem for soil microbiome analyses, we anticipate that plant root and rhizosphere microbiota profiles are likely less affected by relic DNA, as plants continuously secrete exudates from their roots and thereby attract living and exudate-metabolizing microbes. We now mention the limitation associated with relic DNA in the discussion (Line 209).

Reviewer #4 (Remarks to the Author) [Recruited to comment on behalf of Reviewer #2]:

Hu et al., Root exudate metabolites drive plant-soil feedbacks on growth and defense by shaping the rhizosphere microbiota

This is an extremely interesting, if somewhat difficult to follow, manuscript that details interactions between maize plants which secrete benzoxazinoids into soil and the microbiota in the rhizosphere. There are very few studies which demonstrate a causal relationship between root exudate chemistry and rhizosphere communities and so the authors are to be supported in their efforts to do so. Furthermore, since maize is an important agronomic

crop, any demonstrated causal relationships could provide a mechanism to improving crop performance and resistance to disease - with great societal benefits, so the work is clearly important. I have no doubt that the work will be of interest to a wide readership of Nature Communications.

The authors have done a thorough job of responding to the comments and criticisms from the first reviews, much of which were concerned with plant-related issues. I do not wish to rehearse these again but instead would like to focus on the analysis of the amplicon sequencing which I feel needs a comprehensive re-evaluation.

We thank the reviewer for the detailed evaluation and the many useful suggestions regarding our microbiota profiling approach. Before going into more detail, we would like to explain our general approach with this manuscript. As the reviewer identified correctly, we only provide a minimal description of microbial diversity in this manuscript. This is because this manuscript focuses on BX-dependent soil feedback effects rather than BX-effects on microbial ecology and diversity. The latter will be described in more detail and substantiated using multiple soil types and multiple BX mutants in a follow up manuscript (see also comments in the last review round). To keep the manuscript accessible to a broad readership, we wanted to keep the description of microbial ecology and the technical jargon to a minimum.

Nevertheless, we have now conducted all the additional analyses requested by the reviewer (alpha-diversity, pairwise tests, PERMDISP, validation of CAP models) and the revised version of the manuscript now contains the results of these analyses together with a description of microbial diversity effects in the supplement (see also detailed comments below). The Database S3 contains separate markdown reports for bacteria and fungi, where we justify and explain the logic of our data analysis and describe additional results.

The additional analyses support our main findings without affecting the conclusions of the study.

To start at the beginning, the authors compare the rhizosphere communities which develop in association with BX+ve and BX-ve maize lines and bulk soil communities by generating bacterial and fungal amplicon datasets. Having established these sequence datasets, they then jump straight in to comparing β -, between sample, diversity using multivariate ordination and permutation multivariate analysis of variance (PERMANOVA).

- I would suggest that it is necessary to compare the α -, within sample, diversity first since the central hypothesis of the study is that BXs alter the root-associated microbial communities. One logical way to examine this is to test whether species richness (or some other measure of α -diversity) is significantly different between BX+ve and BX-ve. This is a trivial test to perform and differences between the measures should be testable using robust parametric statistics such as analysis of variance and post-hoc tests, if there is indeed a significant effect of treatment. The results of this would be informative since one might hypothesize that if BXs act to either attract specific organisms to the rhizosphere or reduce the rhizosphere community due to toxicity effects then the species richness of BX+ve plants may be reduced. My own preference would be to use Chao-1 estimates of richness, but there are plenty of other measures. Why were α -diversity measures not considered?

We have now performed the requested α -diversity analyses using Chao estimates of richness, Shannon diversity and Pielou's evenness indices. In brief, bacterial and fungal

diversity differs between sample types (soil, rhizosphere and root) but not plant genotype (B73 or bx1(B73)) or soil conditioning (BX+ or BX-) for the field and feedback experiments. The details of the α -diversity analyses can be found in the markdown reports for the bacteria and the fungi in Database S3 and we refer to the in depth analysis of microbial diversity in the main text (Line 89). We also mention that α -diversity was not affected by plant genotype (Line 86).

- The authors next use PERMANOVA to test differences in community structure and show a significant difference between the treatments (bulk soil – WT root – bx1 root – WT rhizosphere – bx1 rhizosphere). There are two issues here: a significant difference observed in PERMANOVA may arise from differences in the dispersion of the data and this can be tested, but wasn't, using the PERMDISP test – I recommend that it be performed here using the “betadisper” code in vegan; secondly, although the PERMANOVA test indicates a significant treatment effect, there are no post-hoc pairwise comparisons of the treatments, the only significant difference could well be the difference between the bulk soil treatment and all the other treatments (it certainly looks possible from the ordinations in Figure 1) - there needs to be some pairwise comparison of the treatment effects.

We have performed the requested pairwise PERMANOVA and dispersion tests. Pairwise testing confirmed the factorial PERMANOVA with the bacteria communities differing significantly between BX exuding and mutant plants while fungi differ only between sample types. PERMDISP additionally revealed that data dispersion significantly differed between soil and rhizosphere samples, which is a finding that is not relevant to the research question of BX exudation. Pairwise testing also confirmed that bacteria communities differed significantly between BX+ and BX- soils in both sample types (and this is not due to differences in dispersion as revealed by PERMDISP) while fungi differed between sample types but not BX soil conditioning. The details of the pairwise PERMANOVA and dispersion analyses can be found in the markdown reports for the bacteria and the fungi in Database S3 and we refer to the in depth analysis of microbial diversity in the main text (Line 89).

- Both unconstrained and constrained ordination procedures were used to visualize differences between the communities developed under each treatment. Bray-Curtis distance metrics were used, but there is no justification regarding why. Were other distance metrics used and was the ordination consistent? Why were weighted Unifrac distances not considered, since they are designed for exactly the type of data generated in this study and the paper by Weiss et al. (Microbiome, 2017, 5:27) - which the authors cite - demonstrates that the metric is both accurate and not subject to sequencing depth-related artefacts (and thus not requiring rarefaction of the data). Weighted Unifrac out performs Bray-Curtis consistently.

Phylogenetic trees, required to calculate the Unifrac distances, are problematic to construct for fungal datasets, because the ITS amplicon sequences are heterogeneous in length and therefore cause unreliable alignments. We agree with the reviewer that the Unifrac distance would be desirable, however, we prioritized the use of a consistent analysis approach and we chose Bray-Curtis as it permits to analyze both bacteria and fungi using the same metric (btw: Bray-Curtis is a “weighted” metric).

- Principal Coordinate Analysis (PCoA) is used to demonstrate differences between the

treatments and both fungal and bacterial communities. In both cases, well less than 50% of the variation in the experiment was accounted for by the first two principal coordinates, which in my experience in soil microbiology is rather low. For both bacteria and fungi, the main difference seems to be between rhizoplane-, rhizosphere-associated communities and the bulk soil since these differences are ordinated along Axis 1 in both cases. This rhizosphere effect is consistent with a great number of studies and not novel. In the case of bacteria, the rhizoplane community certainly appears to be different from the rhizosphere and bulk soil communities, but the difference is small. Does this analysis improve if weighted Unifrac distances are used?

We ran the PCoA using weighted Unifrac distances for the bacteria dataset. As the reviewer correctly assumed, the percentage of variation partitioned by the first two PCo increased using the weighted Unifrac distance (Figure 1). Although, weighted Unifrac would improve the amount of accounted variation by the first two PCo for the bacteria dataset, it would cause an inconsistency in the manuscript with the fungal data (see comment above). As explained above, we prioritize the use of a consistent analysis approach and therefore, we did not implement this suggestion in the manuscript. Furthermore, also PCo analyses that use weighted Unifrac and reveal <50% of variation account for the first two axes are not uncommon (e.g., Lundberg *et al.*, 2012, *Nature*, Fig. 1A with PCo1 39.5%, PCo2 9.4% or Peiffer *et al.* 2013, *PNAS*, Fig. S4 with PCo1 14% and PCo2 8.8%).

- Constrained ordination is then used, although exactly why is never explained. Unconstrained ordination attempts to display the variation of multivariate data in a reduced number of dimensions, while constrained ordination attempts to display the variation that can be explained by measured “environmental factors”. There is very little explanation of how the constrained ordination was implemented – the authors describe using partial canonical analysis of principal coordinates (CAP), but what factors were used to constrain the analysis? There are a number of factors that could be used, including the soil compartment (bulk, rhizosphere, rhizoplane), BX concentrations (or presence/absence) since this is the basis for the hypothesis being tested, the data shown in Supplementary Table 2, or a combination of these. Whatever was used appears to do a poor job of explaining the variation, since the amount is less than that explained by the unconstrained ordinations – raising the questions as to whether the constraining factors were chosen correctly. Also, missing from the constrained ordinations is any test of whether the constraining variables account for a significant proportion of the variability. It is therefore impossible to comment on the validity of the models presented. I must admit that I am less familiar with CAP than the more often encountered redundancy analysis or canonical correspondence analysis, but I am not convinced that the CAP shown here is providing any additional insight into the data

or perhaps even implemented properly. A redundancy analysis is shown in Figure 5, why not apply it here too?

We constrained the CAP ordinations by nesting plant genotype within the factor sample type (~sample type/plant genotype) and by the interaction of the factors sample type and soil conditioning (~ sample type * soil conditioning) for the field and feedback datasets, respectively. This information was previously described in the markdown reports and was now added to the graphs and captions of Figs. 1 and 7.

We based our choice for the constrained ordination method on Ramette (2007, *FEMS Microbiology Ecology*) and the vegan package from Oksanen (2011). We decided not to use redundancy analysis (RDA, similar to principal components analysis) as this is a linear method based on Euclidean distances, which is generally meant to be applied to continuous data. Canonical correspondence analysis (CCA) is based on unimodal species–environment relationships, uses Chi-square distances and the axes can be interpreted as linear combinations of the tested variables. We chose to apply CAP, as it can handle any dissimilarity measure (including Bray-Curtis, we wanted consistency with the other analyses) and more importantly, because we specifically wanted to test the effects of the variable of BX exudation. For this purpose partial ordination can be used (see Ramette, 2007, *FEMS Microbiology Ecology*) as it permits to test a particular experimental variable after elimination of possible effects due to other (environmental) variables. CAP ordinations differ from an RDA or CCA that the constrained ordinations report only the variation that is attributable to the factors of interest.

The reviewer raises the concern that the constrained ordinations explained less variation than the unconstrained ordinations. The sum of eigenvalues of all ordination axes is often used a measure of total variation in a data set and in general, the eigenvalues of constrained ordination are lower than eigenvalues of unconstrained ordination. This is because eigenvalues of unconstrained ordination are guaranteed to be the largest possible ones. The eigenvalues of constrained axes can be at maximum as large as the unconstrained eigenvalues (then they would not constrain anything). With regard to constrained ordinations graphs, there are two ways of reporting variation: One way is to report the proportion of *total* variation explained by the constrained axes (values are typically lower compared to unconstrained ordination) and the other way is to report how much of the *explained* variation can be concentrated on these axes (The latter does not refer to the proportion of total variation!). It is common that values documenting the *explained* variation are higher than the proportions of the *total* variation. To clarify that we report the proportion of *total* variation with the constrained axes, we added this information to the captions of Figs. 1 and 7.

Further, we validated the significance of the models using a permutation test (ANOVA). We now report in each “CAP”-figure with a header the amount of variation that is explained by the constrained variables together with the statistic if it accounts for a *significant* proportion of variability. The details were documented also in the markdown reports.

- The authors describe an interesting complementation experiment where microflora washed from soil conditioned with either BX+ve or BX-ve plants are added back to sterilized soils in an attempt to demonstrate any long-term influence of the established microbiomes on plant metabolism. Redundancy analysis is used here and presents the most compelling evidence in the manuscript for an effect of BXs in the sense that differences between BX+ve

and BX-ve soils are apparent along Axis 1 which is associated with 75% of the variability (considerably more than is the case in the unconstrained analysis presented in Supplementary Figure 7) but again, there is no description of which factors were used to constrain the model and no test of whether the model accounts for a significant amount of the total variability. This information must be reported for an informed assessment of the model to be possible.

The redundancy analysis was conducted on physiological leaf measures (Fig. 5). We validated the significance of the model using a permutation test (ANOVA). The model, variance and model significance were added to the figure.

- Furthermore, regarding the “transplanting” of microbiomes from each soil: what efforts were made to ensure that the comparisons are meaningful? For example, were the numbers of viable cells added to sterilized soils the same, or were extracts just added and assumptions made regarding the number and viability of cells? If the number of viable cells is significantly different between the treatments it is not possible to draw conclusions which relate exclusively to the community composition of the pre-conditioned soils. Nothing fancy here, but a Live-Dead fluorescence assay should help inform this issue. Furthermore, are the extraction procedures properly accounting for any differences in community structure between the different soils? There is no amplicon-based assessment that the differences observed in the soils is replicated by the extracted microflora. The authors make a great many implicit assumptions here for which there is little support.

The reviewer is right to point out that the microbial transplanting experiment, although useful and employed successfully before (Wagg *et al.*, 2014, *PNAS*), generates a number of follow up questions. What can be concluded from this experiment is that transferring a filtered wash that excludes larger members of the soil fauna, but includes smaller members such as bacteria and fungi, is sufficient to recover the benzoxazinoid soil conditioning phenotype. Thus, the soil feedback effects are not due to changes in the distribution and abundance of nematodes or arthropods, but smaller organisms and/or their labile metabolites. It is important to mention that the extracts were standardized for soil mass. In our opinion, the exciting aspect about this assay is that it opens up the possibility to characterize the factor(s) that are transmitted from the conditioning phase to the response phase, using approaches as suggested by the reviewer, but also including further approaches such as additional size fractionation, labeling and LC-MS analyses to understand the contribution of transmitted microbial metabolites to the differential establishment of the microbiomes of the next plant generation (see for instance de Agüero *et al.*, 2016, *Science*). Unfortunately, these experiments are beyond the scope of this paper, but we plan to perform them in the context of a follow up study. We have added a paragraph to the discussion of the manuscript to clarify this aspect and avoid the impression that we make unsupported assumptions.

Line 204: “Future experiments are needed to clarify the exact nature of the BX-dependent factors that are transmitted from the conditioning to the response phase. Complementation experiments suggests that larger members of the soil fauna are not required for the feedback effects (Wagg *et al.*, 2014, *PNAS*). Instead, microbes and/or their metabolites are likely to be transmitted and may determine the assembly of the microbiota in the next generation (de Agüero *et al.*, 2016, *Science*). Detailed microbiome analyses, including approaches that reduce the noise generated by relic DNA in the soil (Carini *et al.*, 2016, *Nature Microbiology*) as well as high-resolution metabolite fingerprinting combined with

activity screens of the microbial extracts from BX+ and BX- conditioned soils will help to test these hypotheses.”

- In the Methods section (page 26, lines 557-559) the authors describe applying rarefaction to their datasets and reference the Weiss et al. paper. Careful reading of the paper suggests that rarefaction is only necessary if sequence reads differ by more than a factor of 10 between samples. Was this the case? Some detail regarding the depth of sequencing between the different datasets is necessary.

For the all different datasets, we documented the mean sequencing depths of the different sample groups in the markdown reports for the bacteria and the fungi in Database S3.

Weiss *et al.* recommend to rarefy the data if the sequencing depths differ significantly between treatment groups. Following Weiss et al., we tested for differences in sequencing depths between the sample groups utilizing the non-parametric Kruskal-Wallis Test. We did not find significant differences in sequence numbers between the sample groups in the fungal dataset. We still decided to employ rarefaction for normalization of the fungal data because we wanted to treat the ITS data the same way as for the 16S data (where we found significant differences in sequence numbers between the sample groups). These details were documented in the markdown reports.

- Why were Cyanobacteria omitted from the analyses? Are they considered not to be part of the soil microbiome?

The 16S rRNA gene data was generated among other sample types from root samples presenting a mixed source of DNA (bacteria and plants). Due to the endosymbiotic origin of plastids, there is high sequence homology between plant plastid and Cyanobacteria 16S rRNA gene sequences. In earlier work applying the PCR primers 799F and 1193R on plant root samples (Schlaeppli *et al.*, 2014, *PNAS*), we noticed that some Cyanobacteria OTUs were specifically enrichment in plant samples and we learned that not all plastid OTUs could be robustly identified based on sequence and taxonomy. Therefore, the removal of the Cyanobacteria presents kind of a ‘safety measure’ to avoid confounding the analysis by the presence of possible plastid sequences.

- Description and justification for the ordination procedures carried out here is necessary. Also, bi-plots of the data and factors used in the constrained ordinations is also necessary, as well as a thorough statistical testing of the models.

As indicated in our replies above, we now describe and justify in more detail the ordination procedures and we have added as well the statistical validation of the models.

References:

- Carini P, Marsden PJ, Leff JW, Morgan EE, Strickland MS, Fierer N. 2016. Relic DNA is abundant in soil and obscures estimates of soil microbial diversity. *Nature Microbiology* 2: 16242.
- de Agüero MG, Ganai-Vonarburg SC, Fuhrer T, Rupp S, Uchimura Y, Li H, Steinert A, Heikenwalder M, Hapfelmeier S, Sauer U. 2016. The maternal microbiota drives early postnatal innate immune development. *Science* 351: 1296-1302.

- Lundberg DS, Lebeis SL, Paredes SH, Yourstone S, Gehring J, Malfatti S, Tremblay J, Engelbrektson A, Kunin V, del Rio TG, et al. 2012.** Defining the core *Arabidopsis thaliana* root microbiome. *Nature* **488**: 86-90.
- Oksanen J 2011.** Multivariate analysis of ecological communities in R: vegan tutorial. R package version: R Foundation for Statistical Computing Vienna, Austria.
- Peiffer JA, Spor A, Koren O, Jin Z, Tringe SG, Dangl JL, Buckler ES, Ley RE. 2013.** Diversity and heritability of the maize rhizosphere microbiome under field conditions. *Proceedings of the National Academy of Sciences of the United States of America* **110**: 6548-6553.
- Ramette A. 2007.** Multivariate analyses in microbial ecology. *FEMS Microbiology Ecology* **62**: 142-160.
- Schlaeppli K, Dombrowski N, Oter RG, Ver Loren van Themaat E, Schulze-Lefert P. 2014.** Quantitative divergence of the bacterial root microbiota in *Arabidopsis thaliana* relatives. *Proceedings of the National Academy of Sciences of the United States of America* **111**: 585-592.
- Wagg C, Bender SF, Widmer F, van der Heijden MG. 2014.** Soil biodiversity and soil community composition determine ecosystem multifunctionality. *Proceedings of the National Academy of Sciences of the United States of America* **111**: 5266-5270.

REVIEWERS' COMMENTS:

Reviewer #4 (Remarks to the Author):

The authors have addressed the issues raised in review comprehensively and clearly. The additions they have made to the manuscript have added important clarity and their justifications for the approaches adopted are reasonable. I am happy that this manuscript is ready for acceptance.